# Autism-associated *SHANK3* missense point mutations impact conformational fluctuations and protein turnover at synapses

Michael Bucher[1,2,3], Stephan Niebling[4], Yuhao Han[2,5], Dmitry Molodenskiy[6], Fatemeh Hassani Nia[7], Hans-Jürgen Kreienkamp[7], Dmitri Svergun[6], Eunjoon Kim[8], Alla S Kostyukova[2,9], Michael R Kreutz[3,10,11,12]*, Marina Mikhaylova[1,2]*

[1]AG Optobiology, Institute of Biology, Humboldt-University, Berlin, Germany; [2]DFG Emmy Noether Guest Group 'Neuronal Protein Transport', Institute for Molecular Neurogenetics, Center for Molecular Neurobiology (ZMNH), University Medical Center Hamburg-Eppendorf (UKE), Hamburg, Germany; [3]RG Neuroplasticity, Leibniz-Institute for Neurobiology (LIN), Magdeburg, Germany; [4]Molecular Biophysics and High-Throughput Crystallization, European Molecular Biology Laboratory (EMBL), Hamburg, Germany; [5]Structural Cell Biology of Viruses, Centre for Structural Systems Biology (CSSB) and Leibniz Institute for Experimental Virology, Hamburg, Germany; [6]European Molecular Biology Laboratory (EMBL) Hamburg Unit, DESY, Hamburg, Germany; [7]Institute of Human Genetics, Center for Obstetrics and Pediatrics, University Medical Center Hamburg-Eppendorf (UKE), Hamburg, Germany; [8]Center for Synaptic Brain Dysfunctions, Institute for Basic Science (IBS) and Department of Biological Sciences, Korea Advanced Institute of Science and Technology (KAIST), Daejeon, Republic of Korea; [9]The Gene and Linda Voiland School of Chemical Engineering and Bioengineering, Washington State University (WSU), Pullman, United States; [10]Leibniz Group 'Dendritic Organelles and Synaptic Function', Center for Molecular Neurobiology (ZMNH), University Medical Center Hamburg-Eppendorf (UKE), Hamburg, Germany; [11]German Center for Neurodegenerative Diseases, Magdeburg, Germany; [12]Center for Behavioral Brain Sciences, Magdeburg, Germany

*For correspondence:
michael.kreutz@zmnh.uni-hamburg.de (MRK);
marina.mikhaylova@hu-berlin.de (MM)

Competing interests: The authors declare that no competing interests exist.

**Abstract** Members of the SH3- and ankyrin repeat (SHANK) protein family are considered as master scaffolds of the postsynaptic density of glutamatergic synapses. Several missense mutations within the canonical SHANK3 isoform have been proposed as causative for the development of autism spectrum disorders (ASDs). However, there is a surprising paucity of data linking missense mutation-induced changes in protein structure and dynamics to the occurrence of ASD-related synaptic phenotypes. In this proof-of-principle study, we focus on two ASD-associated point mutations, both located within the same domain of SHANK3 and demonstrate that both mutant proteins indeed show distinct changes in secondary and tertiary structure as well as higher conformational fluctuations. Local and distal structural disturbances result in altered synaptic targeting and changes of protein turnover at synaptic sites in rat primary hippocampal neurons.

## Introduction

De novo and inherited point mutations contribute to several neuropsychiatric disorders and are common in genes that are responsible for synaptic function (*Gratten et al., 2013*; *Hammer et al., 2015*; *Penzes et al., 2011*). Currently, very little is known about how such missense mutations alter protein structure and conformational stability in a manner that might cause disease-related synaptic phenotypes. Establishing such structure-function relationships will contribute to a better understanding of pathogenic mechanisms in particular for autism spectrum disorder (ASD; *Luo et al., 2018*; *Stefl et al., 2013*).

SHANK3 is a multidomain synaptic scaffold protein most prominently expressed in the brain (*Grabrucker et al., 2011*). To date, multiple splice isoforms of SHANK3 with varying domain organization have been identified (*Wang et al., 2014a*). The SHANK3a isoform shows highest expression in the striatum and hippocampus and consists of five distinct domains plus an additional proline-rich cluster (*Jiang and Ehlers, 2013*; *Wang et al., 2014b*). These include the SHANK/ProSAP N-terminal (SPN) domain followed by the ankyrin repeat region (ARR), a Src homology 3 (SH3) domain, the PSD-95/DLG/ZO-1 (PDZ) domain, and the C-terminal sterile alpha motif (SAM). The SPN domain has been shown to interact with small GTPases of the Ras superfamily including R-Ras, H-Ras or Rap1, which are involved in the regulation of synaptic F-actin structure and dynamics and in postsynaptic signal transduction (*Cai et al., 2020*; *Lilja et al., 2017*). The ARR domain binds the cytoskeletal protein α-fodrin, an adhesive junction associated protein δ-catenin, and a component of a ubiquitin ligase complex, sharpin (*Böckers et al., 2001*; *Lim et al., 2001*; *Hassani Nia et al., 2020*). While the SH3 domain directly associates with the $Ca^{2+}$ channel $Ca_v1.3$, the PDZ domain is involved in a direct interaction with synapse-associated protein 90/postsynaptic density-95-associated protein 1 (SAPAP1) or the GluA1 subunit of α-amino-3-hydroxy-5-methyl-4-isoxazole propionic acid (AMPA) receptors (*Boeckers et al., 1999*; *Uchino et al., 2006*). Between the PDZ domain and the C-terminal SAM domain, SHANK3 contains a proline-rich region harboring multiple protein interaction sites including those for homer1 and cortactin, which are relevant for linking of SHANK3 to other PSD scaffolds and the regulation of spinous actin dynamics (*MacGillavry et al., 2016*; *Naisbitt et al., 1999*; *Tu et al., 1999*). Finally, the SAM domain facilitates oligomerization of SHANK3 within the postsynaptic density (PSD) and is required for postsynaptic targeting (*Baron et al., 2006*; *Boeckers et al., 2005*). Thus, SHANK3 acts as 'master organizer' of the PSD via multiple protein interactions and the resulting indirect association with ionotropic glutamate receptors (*Jiang and Ehlers, 2013*; *Zeng et al., 2018*; *Zhang et al., 2005*) and is thus crucial for synaptic structure and function (*Grabrucker et al., 2011*; *Monteiro and Feng, 2017*).

Disruption of SHANK3 function has been linked to numerous neuropsychiatric and neurodevelopmental disorders (*Durand et al., 2007*; *Gauthier et al., 2009*). In fact, it is one of the few proteins with a clear genetic linkage to synaptic dysfunction in conditions like the Phelan-McDermid syndrome (PMS) and other ASDs disease states collectively coined as shankopathies (*Guilmatre et al., 2014*; *Sala et al., 2015*; *Wang et al., 2014a*). Pathogenic rearrangements in the *SHANK3* gene result in either copy-number or coding-sequence variants, which have been shown to be of high clinical relevance (*Boccuto et al., 2013*; *Leblond et al., 2014*). While alterations in copy-number caused by gene deletions, ring chromosomes, unbalanced translocations or interstitial deletions have been studied extensively in several *SHANK3* knockout mouse models (*Bozdagi et al., 2010*; *Peixoto et al., 2016*; *Qin et al., 2018*; *Yi et al., 2016*; *Yoo et al., 2019*), the impact of deleterious *SHANK3* coding-sequence variants on protein structure and a corresponding synaptic phenotype is much less clear. Only one study reported the generation of a *SHANK3* knock-in mouse line carrying the Q321R mutation identified in a human individual with ASD. Homozygous knock-in mice showed reduced levels of SHANK3a, suggesting an altered protein stability and the physiological and behavioral characterization revealed decreased neuronal excitability, repetitive and anxiety-like behavior, EEG patterns, and seizure susceptibility (*Yoo et al., 2019*). Single nucleotide deletions or insertions can lead to frameshifts, premature stop codons, or changes in splicing whereas more frequently occurring missense mutations could have an impact on the local or global protein structure (*Hassani Nia and Kreienkamp, 2018*).

The crystal structure for an N-terminal SHANK3 fragment comprising the SPN and ARR domain has recently become available (PDB 5G4X; *Lilja et al., 2017*). Two autism-related point mutations, R12C and L68P, have been described in patients and are located within this region (*Hassani Nia and*

*Kreienkamp, 2018*) and therefore represent an attractive target for structural analysis. In previous work, the L68P mutation has been shown to alter G-protein signaling and integrin activation as well as to result in enhanced binding to protein ligands such as α-fodrin or sharpin, whereas the R12C mutation has been reported to have moderate effects on spine formation and synaptic transmission (*Durand et al., 2012*; *Lilja et al., 2017*; *Mameza et al., 2013*). The R12C mutation was originally identified in an autistic patient suffering from severe mental retardation and total absence of language, who inherited the mutation from his mother (*Durand et al., 2007*). The L68P mutation was transmitted by an epileptic father and was shown to result in language disorder and ASD in a female patient (*Gauthier et al., 2009*).

In this study, using a wide range of biophysical and cellular approaches we show that the ASD-associated point mutations R12C and L68P affect different levels of protein structure. While the R12C mutation confers increased secondary structure stability and reduces synaptic residing time of SHANK3, the L68P mutation results in partial unfolding with reduced tertiary structure stability and an increased number of dendritic SHANK3 clusters. Thus, subtle mutation-induced changes in tertiary structure come along with altered conformational fluctuations that will likely cause ASD-related synaptic phenotypes.

## Results

### SHANK3 L68P and R12C mutants show altered folding and complex topology

We first aimed to provide structural underpinnings that might be causally linked to the pathological role of the two inherited ASD-associated missense mutations located within the SPN domain of SHANK3 (*Figure 1—figure supplement 1*). To that end, we examined the low-resolution structure of a larger SHANK3 fragment covering amino acids 1 to 676 including the SPN, ARR, SH3, and PDZ domains in solution by small-angle X-ray scattering (SAXS). This method is especially powerful to analyze structural changes of large and conformationally flexible multidomain proteins in solution (*Blanchet and Svergun, 2013*). To purify each individual SHANK3$^{(1-676)}$ fragment (a wild type/WT, R12C-, and L68P-mutant), we fused an N-terminal His$_6$-SUMO-tag to enable protein purification via Ni$^{2+}$ affinity chromatography (*Figure 1—figure supplement 2A*). Eluted proteins were then further purified by size-exclusion chromatography (SEC, *Figure 1A*). Notably, for SAXS we did not cleave the tag in order to improve solubility and to allow for measurements over a broader concentration range. We found that both the WT and R12C mutant showed a linear dependence of the radius of gyration (R$_g$) on protein concentration. For the L68P mutant, R$_g$ also increased with concentration but showed no linear dependency, which might indicate increased interparticle interactions due to partial unfolding (*Figure 1B*). Additionally, we observed an increase of the maximum particle diameter (D$_{max}$) with increasing protein concentration from pair distance distribution functions (PDDFs), which were calculated from SAXS profiles measured for each His$_6$-SUMO-SHANK3$^{(1-676)}$ variant at indicated concentrations (*Figure 1—figure supplement 3B*). An overview of principle parameters calculated from SAXS data is provided in *Supplementary file 1*.

Taken together, these data suggest attractive interparticle interactions in solution independent of the SAM domain. To analyze whether His$_6$-SUMO-SHANK3$^{(1-676)}$ variants could exist in an oligomeric form in solution even without the presence of the SAM domain, SAXS profiles measured for the highest and lowest protein concentration were merged and fitted with *CORAL* using a dimeric symmetry constraint. Interestingly, we found that all three SHANK3 variants can form dimers via their SH3 and PDZ domains in a 2 × 2 stoichiometry (*Figure 1—figure supplement 4*).

However, the fitted curve for the dimer deviates from experimental data in the very low angular range, suggesting the existence of a monomer-dimer equilibrium. Independently, *OLIGOMER* analysis of SAXS data indicated a linear decrease of the monomeric SHANK3 volume fraction with increasing concentration, thereby further corroborating a dynamic concentration-dependent monomer-dimer equilibrium (*Figure 1—figure supplement 3C*). While the R12C mutant did not show any significant changes in the dimeric protein complex topology compared to the WT, the relative orientation of the ARR domain was altered with respect to the [(SH3)-(PDZ)]$_2$ cluster for the L68P mutant. This indirectly suggests that not only the SAM domain but also the SH3 and PDZ domains might be actively involved in the formation of an oligomeric SHANK3 scaffold in the PSD. Consequently, it

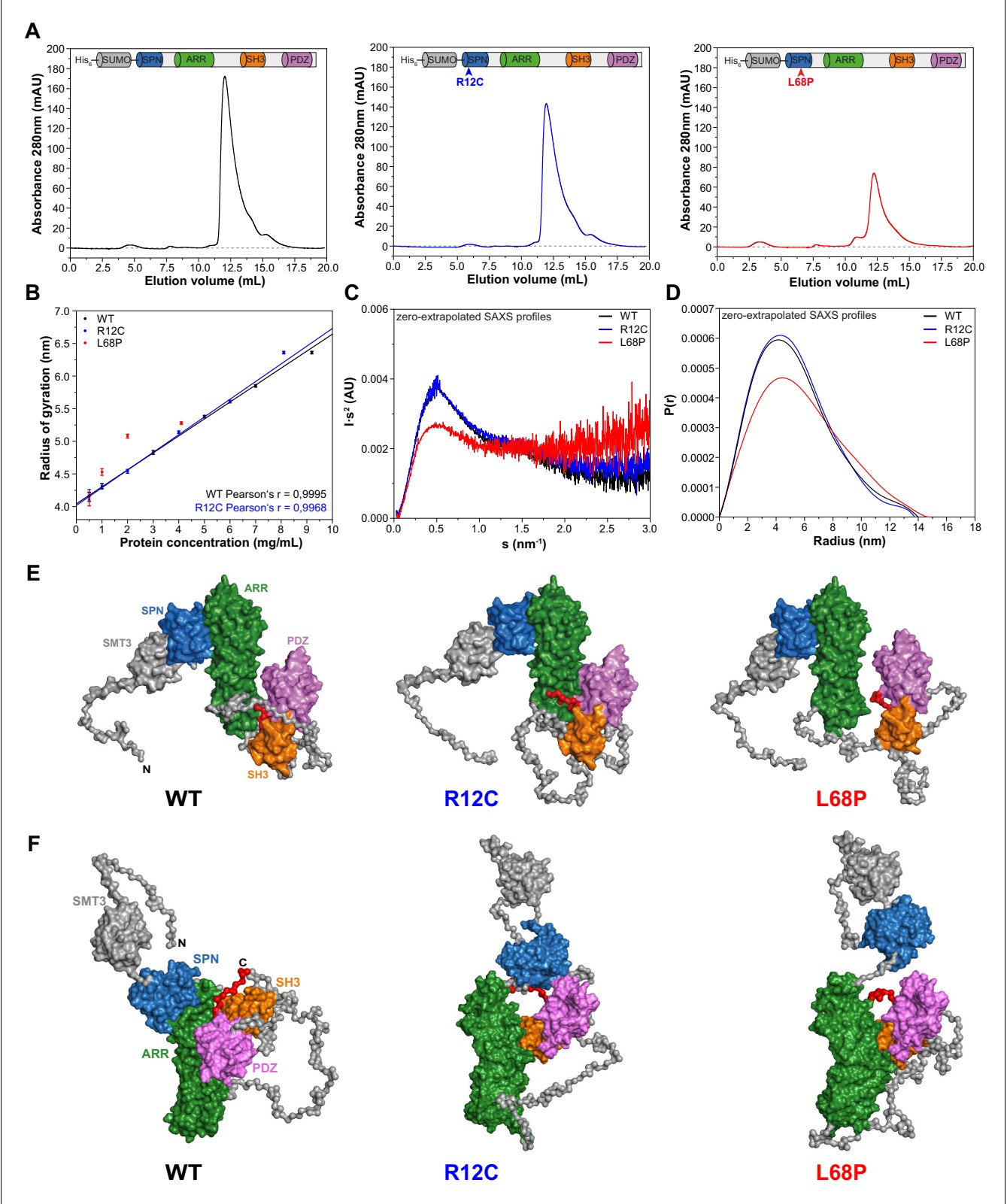

**Figure 1.** Small-angle X-ray scattering from ASD-associated SHANK3 mutants shows changes in protein folding and topology. (**A**) Size-exclusion chromatograms of Ni²⁺-IDA pre-purified His₆-SUMO-SHANK3⁽¹⁻⁶⁷⁶⁾ variants are shown (elution peak at ~12.0 mL). The individual elution peaks were used for SAXS. (**B**) $R_g$ values derived from Guinier approximation in the range of $sR_g$ <1.3 of SAXS profiles measured at different protein concentrations. The WT and R12C mutant show a linear increase of $R_g$ with protein concentration, which suggests the presence of attractive interparticle interactions. *Figure 1 continued on next page*

*Figure 1 continued*

(C) Kratky plots from zero-extrapolated SAXS profiles of the WT and R12C mutant resemble the profile of a folded multidomain protein with flexible linkers, while the L68P mutant appears to be partially unfolded. (D) Distance distribution curves were computed with GNOM using zero-extrapolated SAXS profiles as input data. Particles were assumed to be arbitrary monodisperse. The curves indicate a maximum particle diameter ($D_{max}$) of ~14–15 nm for monomeric SHANK3$^{(1-676)}$ without significant alterations due to ASD-associated mutations. (E) Rigid-body *CORAL* models of the SHANK3 complex topology in solution. High-resolution structures of individual SHANK3 fragments were fitted against zero-extrapolated SAXS profiles without assuming any higher order symmetry (space group P1). The SPN and ARR domains were treated as a single rigid body (PDB 5G4X). The models indicate distal effects of both mutations on the position of the SH3 domain (orange) and PDZ domain (magenta) relative to the ARR domain (green) as well as changes in the orientation of linker regions. (F) *CORAL* models of monomeric His$_6$-SUMO-SHANK3$^{(1-676)}$ variants with split SPN/ARR domains, where the linker region has been replaced by flexible dummy residues, capture mutation-induced perturbations of the SPN/ARR domain interface. Perspectives of visualized structures were chosen to facilitate highest visibility of structural regions of interest. All acquired SAXS data including fits and models were deposited to SASBDB (WT: SASDLJ3, R12C: SASDLL3 and L68P: SASDLK3). SAXS = small-angle X-ray scattering, ASD = autism spectrum disorders, $R_g$ = radius of gyration, WT = wild type, SPN = SHANK/ProSAP N-terminal, ARR = ankyrin repeat.

The online version of this article includes the following figure supplement(s) for figure 1:

**Figure supplement 1.** Localization of ASD-associated mutation sites in the SHANK3$^{(1-346)}$ input topology of molecular dynamics simulations.

**Figure supplement 2.** Schematic overview of protein purification steps involved in the preparation of His$_6$-SUMO-SHANK3$^{(1-676)}$ and SHANK3$^{(1-676)}$ variants.

**Figure supplement 3.** Guinier plots and pair distance distribution functions (PDDFs) from SAXS profiles measured at different concentrations.

**Figure supplement 4.** CORAL-derived models of a dimeric SHANK3 protein complex topology in solution.

**Figure supplement 5.** Zero-extrapolated SAXS data fitted with CORAL to derive structural models of monomeric His$_6$-SUMO-SHANK3$^{(1-676)}$ variants in solution.

would be of great value to obtain detailed information about the concentration of SHANK3 at the synapse as this would allow an estimation of the contribution of SH3 and PDZ domain-mediated dimerization in the spine.

To facilitate further structural analyses, however, these interparticle effects were removed by extrapolation to infinite dilution. Since the estimated monomeric SHANK3 volume fraction at zero concentration is close or equal to unity for all variants, zero-extrapolated SAXS profiles represent the monomeric state of His$_6$-SUMO-SHANK3$^{(1-676)}$. For the WT and R12C mutant, Kratky plots generated from zero-extrapolated SAXS profiles had the shape of a folded multidomain protein with a flexible linker (*Figure 1C*). In contrast, the L68P mutant showed the characteristic shape of a partially unfolded protein. Furthermore, PDDFs revealed that both the WT and mutants exhibit monomeric $D_{max}$ values of ~14–15 nm, consistent with values obtained from low-concentration SAXS profiles (*Figure 1D*, *Figure 1—figure supplement 3B*). Finally, *CORAL* fitting of zero-extrapolated SAXS profiles without any symmetry constraint (space group P1) showed additional structural differences in the corresponding monomers (*Figure 1E and F*, *Figure 1—figure supplement 5*). Interestingly, for both mutants the modeled monomeric protein structure exhibited a change in the relative position of the SH3 and PDZ domains, which are in turn separated from the ARR domain by an intrinsically disordered region of 122 amino acids (*Figure 1E*). While, in the R12C mutant, the SH3 and PDZ domains seem to be connected to the ARR domain more strongly, the L68P mutant shows a decoupling of these domains. To additionally capture potential mutation-induced disruptions of the interface between the SPN and ARR domain, we performed *CORAL* fits, where the linker region of the SPN-ARR fragment (PDB: 5G4X, *Lilja et al., 2017*) has been replaced by flexible dummy residues (DRs). Expectedly, this resulted in an improvement of fits due to the increased degrees of freedom and revealed that the SPN domain is decoupled from the ARR domain in both mutants, indeed suggesting a disruptive effect of both mutations on the domain interface (*Figure 1F*). Collectively, the data indicate that the ASD-associated mutations R12C and L68P result in altered protein folding (L68P) and complex topology in solution.

## The L68P mutation results in reduced tertiary structure stability

Since the SAXS data suggest changes in the tertiary structure of ASD-associated SHANK3 mutants, we tested this hypothesis with nano-differential scanning fluorimetry (nDSF; *Alexander et al., 2014*). To exclude a possible contribution of His$_6$-SUMO to thermal transitions, we removed the tag by treatment with Sentrin-specific protease 2 (SenP2; *Figure 1—figure supplement 2B and C*) and obtained individual SHANK3$^{(1-676)}$ variants in sufficient purity for nDSF (*Figure 2B*). We monitored the thermal unfolding of purified SHANK3$^{(1-676)}$ variants label-free using intrinsic tryptophan

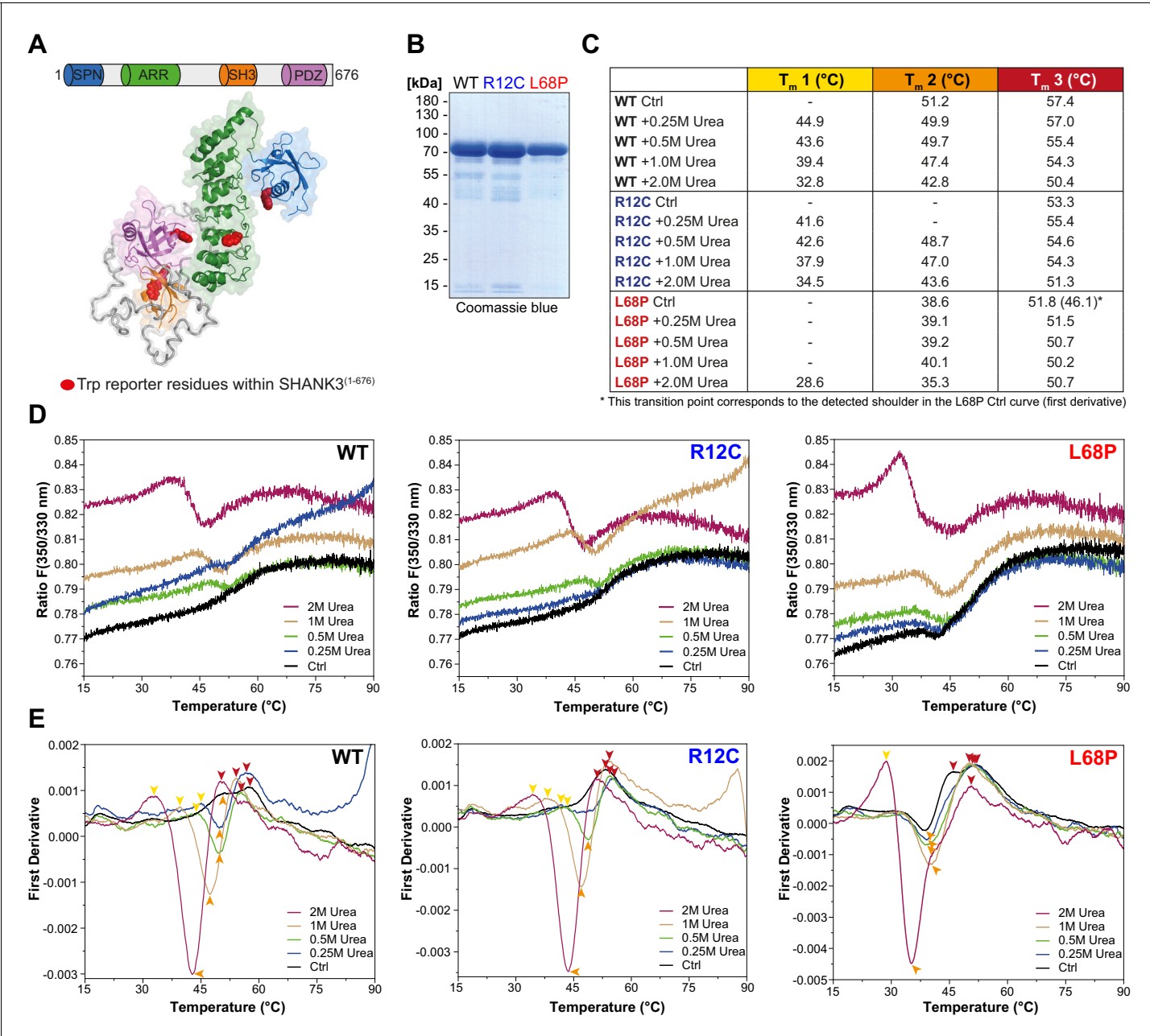

**Figure 2.** ASD-associated SHANK3 mutations differentially affect protein tertiary structure. (**A**) Schematic representation of the SHANK3$^{(1-676)}$ fragment, which was used for nDSF measurements. Intrinsic tryptophan (Trp) reporter residues are highlighted in the structure, which was derived from SAXS data. (**B**) SDS-PAGE of Ni$^{2+}$-NTA purified SHANK3$^{(1-676)}$ variants which were used for nDSF and CD spectroscopy. (**C**) Overview of detected melting points from peaks of first derivative curves (from E). Due to the complex melting behavior, melting was classified in three transition zones (Tm1–Tm3) which are partially overlapping. (**D**) Label-free determination of thermal and chemical stability of purified SHANK3$^{(1-676)}$ variants by intrinsic fluorescence emission depicted as ratio of 350/330 nm as a function of temperature. Melting curves were acquired at a protein concentration of ~0.5 mg/mL, 50% excitation power and with a heating rate of 1°C/min. (**E**) First derivative analysis of melting curves shown in (**D**). Transition points are indicated with colored arrowheads (color-coded according to the transition zones) and shifted toward lower temperatures with increasing urea concentration, as expected. For the L68P mutant, considerably lower melting points are detected compared to the WT or R12C mutant, suggesting a reduced thermal stability of the tertiary structure. ASD = autism spectrum disorders, SAXS = small-angle X-ray scattering, WT = wild type.

The online version of this article includes the following source data and figure supplement(s) for figure 2:

**Source data 1.** ASD-associated SHANK3 mutations differentially affect protein tertiary structure.

**Figure supplement 1.** Intrinsic tryptophan fluorescence emission spectra of His$_6$-SUMO-SHANK3$^{(1-676)}$ variants.

**Figure supplement 2.** Measurement of His$_6$-SUMO-SHANK3$^{(1-676)}$ surface hydrophobicity by extrinsic ANS fluorescence spectroscopy.

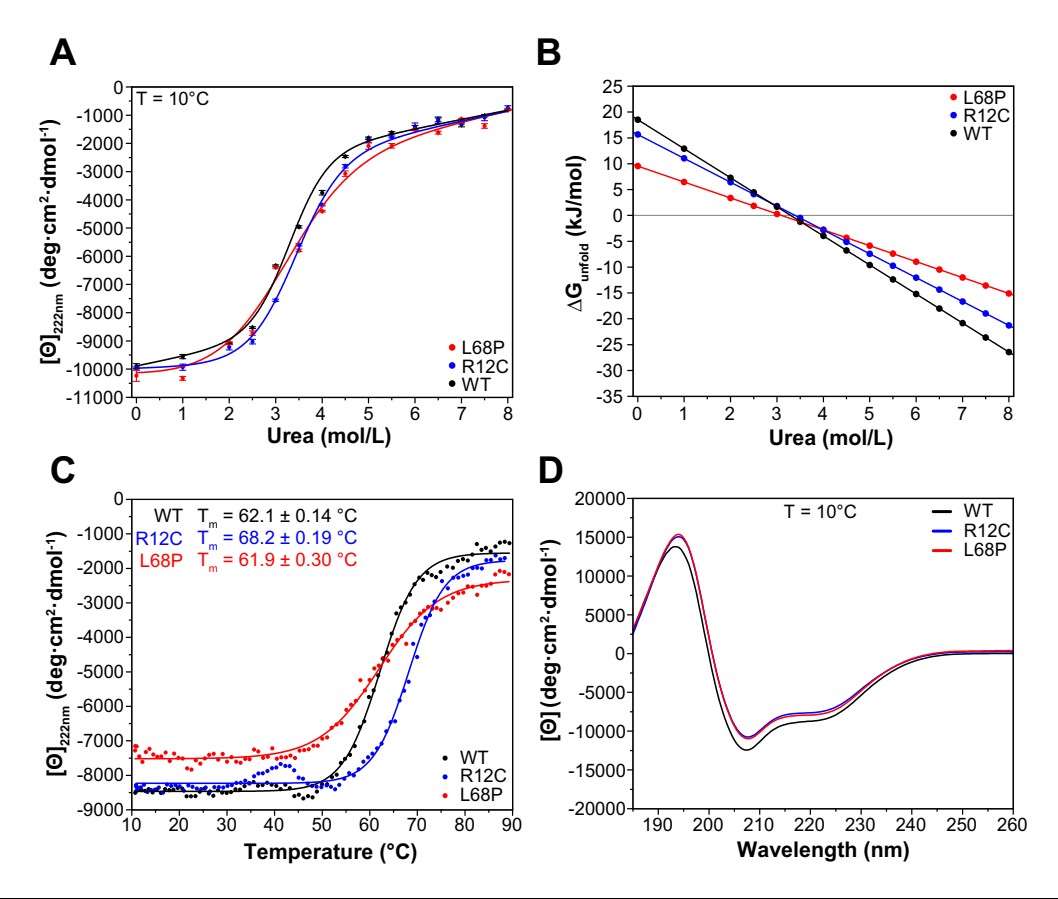

**Figure 3.** ASD-associated SHANK3 mutants show structural alterations at secondary structure level. (**A**) Isothermal chemical unfolding curves with urea as denaturant are depicted (222 nm, 3 s/point, 25 μs sample period, three repeats, T = 10.0 ± 0.2°C, protein concentration ~0.15 mg/mL). The data has been fitted to a two-state unfolding transition model. Transition midpoints do not vary significantly but differences occur in the slopes of the corresponding folded and unfolded baselines. (**B**) Unfolding free energies are calculated from fitted parameters obtained in (**A**) using a linear free energy model. Both mutants show a reduced unfolding cooperativity and thermodynamic stability when unfolding is induced with urea. (**C**) Thermal unfolding curves have been acquired from samples measured in (**B**) in the presence of 2.0 M urea to prevent thermally induced protein aggregation preceding actual secondary structure melting (222 nm, 3 s/point, 25 μs sample period, 10–90°C T range, 1.0°C step size, 1.0°C/min stepped ramping with 30 s settling time and 0.5°C tolerance). Thermal unfolding curves reveal an increased secondary structure stability of the R12C mutant (dynode voltage = 370–400 V between 10 and 90°C). (**D**) Far-UV CD spectra of SHANK3[(1-676)] variants are shown in the range of 260–185 nm (0.5 nm step size, 3 s/point, 25 μs sample period, three repeats, T = 10.0 ± 0.2°C). Immediately before measurement, proteins have been buffer exchanged to 10 mM $KH_2PO_4$, 100 mM KF, 0.5 mM DTT, pH = 6.50. Differences between WT and mutant protein spectra are minor. ASD = autism spectrum disorders, WT = wild type.

The online version of this article includes the following source data and figure supplement(s) for figure 3:

**Source data 1.** Chemical and thermal unfolding.
**Source data 2.** Thermodynamic stability of SHANK3 proteins.
**Source data 3.** Far-UV CD spectra of SHANK3 proteins.
**Figure supplement 1.** Limited trypsin proteolysis of SHANK3[(1-676)] variants.
**Figure supplement 2.** CD PMT voltage as a function of wavelength.

fluorescence emission of the untagged protein (*Figure 2A and B*). To suppress precipitation of melted proteins, we added increasing concentrations of urea (0.25–2.0M) to the samples. Although aggregation was largely prevented by addition of 2.0M urea, protein precipitates were still visible after completing thermal denaturation runs, indicating the irreversibility of thermal unfolding, which is known for many large multidomain proteins (*Strucksberg et al., 2007*). Therefore, we compared the thermal transitions on a qualitative level and classified the complex melting behavior into three partially overlapping thermal transition zones ($T_m1$–$T_m3$; *Figure 2C and E*). Individual transition points have been identified by peak detection of first derivative curves, which were determined from the corresponding fluorescence ratio curves (F 350/330 nm vs. T; *Figure 2D and E*). Interestingly, we found only a single transition point for the R12C mutant in the absence of urea, while both the WT and L68P mutant showed multiple transitions (*Figure 2E*). However, upon titration with urea the L68P mutant clearly showed a strongly increased structural susceptibility as we observed a pronounced shift of transition zones 1 and 2 toward lower temperatures (~4–6°C in $T_m1$ and ~8°C in $T_m2$). This shift was essentially absent for the R12C mutant, which displayed transition temperatures very similar to the WT. To independently verify the reduced tertiary structure stability of the L68P mutant, we performed intrinsic tryptophan fluorescence spectroscopy measurements using $His_6$-SUMO-SHANK3$^{(1-676)}$ variants, which have also been used for SAXS before. Consistently, fluorescence spectra showed a marked reduction in the peak intensity of fluorescence emission between 330 and 350 nm for the L68P mutant, indicative of partial unfolding (*Figure 2—figure supplement 1*). Furthermore, by extrinsic fluorescence emission spectroscopy we observed an approximately twofold increase in the peak fluorescence intensity of the dye 1-anilinonaphthalene-8-sulphonate (ANS) for the L68P mutant compared to WT and R12C (*Figure 2—figure supplement 2*). Since ANS becomes strongly fluorescent only upon binding to a hydrophobic environment, these measurements indicate an increased surface hydrophobicity of the L68P mutant. Collectively our data reveal that the L68P, but not the R12C mutation, reduces stability and results in perturbed tertiary structure.

## The autism-related SHANK3 missense variants R12C and L68P show changes in secondary structure stability and unfolding cooperativity

To understand how the observed structural deviations from the WT protein such as partial unfolding, altered complex topology in solution, or reduced tertiary structure stability translate to the secondary structure level, we next performed far-ultraviolet circular dichroism (far-UV CD) spectroscopy and monitored both chemical and thermal unfolding of ASD-associated SHANK3 mutants. For direct comparison, identical protein preparations were used for nDSF (*Figure 2B*) and CD spectroscopy experiments. We initially conducted isothermal equilibrium chemical unfolding experiments using urea as denaturant and followed induced structural changes by measuring ellipticities at 222 nm, reporting on α-helical unfolding (*Figure 3A*). We then fitted the data to a two-state unfolding transition model using a linear extrapolation method (LEM) to obtain the free energy of unfolding ($\Delta G_{unfold}$) as a function of denaturant concentration. Interestingly, we found only subtle differences in the transition midpoint concentration of urea ($\Delta G_{unfold}$ = 0 for urea concentration of 3.3 mol/L [WT], 3.4 mol/L [R12C], and 3.1 mol/L [L68P]) but observed a reduction in unfolding cooperativity of both mutants due to a reduced dependence of $\Delta G_{unfold}$ on denaturant concentration (*Figure 3B*). This suggests a mutation-induced destabilization of the SHANK3 secondary structure, which is distinct from the observed tertiary structure perturbations. It has been reported that chemically and thermally denatured protein states may differ substantially. We, therefore, next aimed to complement our chemical unfolding experiments with thermal unfolding measurements (*Narayan et al., 2019*). We observed that the presence of 2.0M urea induced a strong reduction of transition temperatures detected by nDSF but almost no change in ellipticity at 222 nm (*Figures 2E* and *3A*). Therefore, we acquired CD melting curves of SHANK3$^{(1-676)}$ variants in the presence of 2.0M urea from the same sample which was used for chemical unfolding. Strikingly, CD melting temperatures differed substantially from those observed by nDSF (~62–68 vs. ~29–51°C) and showed an increase in the thermal stability of the R12C mutant by approximately 6°C compared to the WT (*Figure 3C*). The L68P mutant, however, did not show an altered melting temperature compared to the WT. In line with this, limited trypsin proteolysis experiments suggest a mildly decreased proteolytic cleavability of the R12C mutant (*Figure 3—figure supplement 1*). Additionally, we did not detect significant discrepancies between the WT and both mutants in their far-UV CD spectra,

suggesting that their secondary structure content is similar (*Figure 3D*). Overall, we observed thermally distinguishable secondary and tertiary structure unfolding as well as differential susceptibility of both mutants to either chemical or thermal perturbations. We conclude that both ASD-associated missense variants show changes in their stability and exhibit reduced chemical unfolding cooperativity.

## Molecular dynamics simulations reveal reduced nanosecond peptide backbone dynamics of both mutants

We next aimed to gain further insights into the conformational dynamics of ASD-associated SHANK3 mutants at very high temporal resolution (*Karplus and Kuriyan, 2005*). Molecular dynamics simulations (MDS) provide a valuable link between protein structure and dynamics (*Hollingsworth and Dror, 2018*). Since both mutations are located within the SPN domain, we took advantage of the published crystal structure of an N-terminal SHANK3 fragment (PDB 5G4X; *Lilja et al., 2017*). To extensively sample the conformational space of SHANK3$^{(1-346)}$ WT, R12C, and L68P, we calculated 1000 ns MD trajectories for each variant. Thereby root mean square fluctuation (RMSF) analysis of C$_\alpha$ atoms revealed an increased conformational flexibility of mutants compared to the WT within the first 100 amino acid residues, corresponding to the SHANK3 SPN domain (*Figure 4A*). Surprisingly, however, root mean square deviation (RMSD) analysis showed that this increase in conformational dynamics seems to be unstable over time. While mutants exhibit higher dynamics during the first 300–400 ns, the WT undergoes a prominent switch after approximately 500 ns and further on consistently shows increased RMSDs compared to the mutants (*Figure 4B*). This switch is essentially absent in both mutants.

To address which domain might contribute to this conformational switch, we performed separate RMSD analyses for the SPN and ARR domains. Indeed, RMSD traces restricted to the SPN domain revealed the same pattern as observed for the whole protein, including a profound increase of RMSD at around 500 ns for the WT (*Figure 4C*). Additionally, this RMSD peak within the SPN domain was absent in both ASD mutants. On the opposite, the point mutations rather induced an increase in the number and amplitude of individual RMSD spikes, suggesting more pronounced structural oscillations of the SPN domain. Interestingly, also within the ARR domain, we observed conformational transitions of the WT as sequential stepwise RMSD increases (*Figure 4D*) and again, this behavior was absent in both mutant proteins (*Figure 4—videos 1–3*). Taken together, MDS identifies the SPN domain as a hotspot of conformational flexibility with higher fluctuations in both ASD-associated mutant proteins (*Figure 4E*). On average, however, both point mutations resulted in reduced backbone dynamics and a lack of discrete conformational transitions, which might be related to structurally regulated protein function.

## SHANK3 mutants form a higher number of dendritic shaft synapses

We next asked whether any consequences will result from the structural perturbations in R12C and L68P SHANK3 mutations in cellular context. Therefore, we performed immunocytochemical labeling of endogenous homer and bassoon as excitatory postsynaptic and presynaptic markers, respectively, and analyzed their co-localization with endogenous SHANK3 (*Figure 5A*). We found that approximately 70% of homer-positive excitatory synapses contained SHANK3 while SHANK3-positive synapses co-localized with homer nearly to 100% (*Figure 5B*). Furthermore, approximately 80% of SHANK3 clusters had a bassoon-positive presynaptic contact. To analyze cellular consequences induced by SHANK3 missense mutations, we overexpressed full-length GFP-tagged SHANK3a variants, which is the most abundant isoform, in primary rat hippocampal neurons for less than 24 hr to prevent protein dosage-dependent effects as higher SHANK3 expression levels directly affect spine structure and function (*Han et al., 2013*; *Roussignol et al., 2005*). Consequently, spine density analysis showed that short-term expression of GFP-SHANK3 variants did not significantly alter spine numbers compared to a GFP-control (*Figure 5C*; *Figure 5—figure supplement 1*). However, neurons expressing the L68P mutant exhibited a significantly increased number of dendritic GFP-SHANK3 L68P clusters in the shaft compared to the WT, suggesting impaired localization to dendritic spines (*Figure 5D*). The R12C mutant showed the same, yet less pronounced trend. Line profile analyses of SHANK3 intensity distributions between spines and dendrites, however, revealed that spines contained on average approximately three times more GFP-SHANK3,

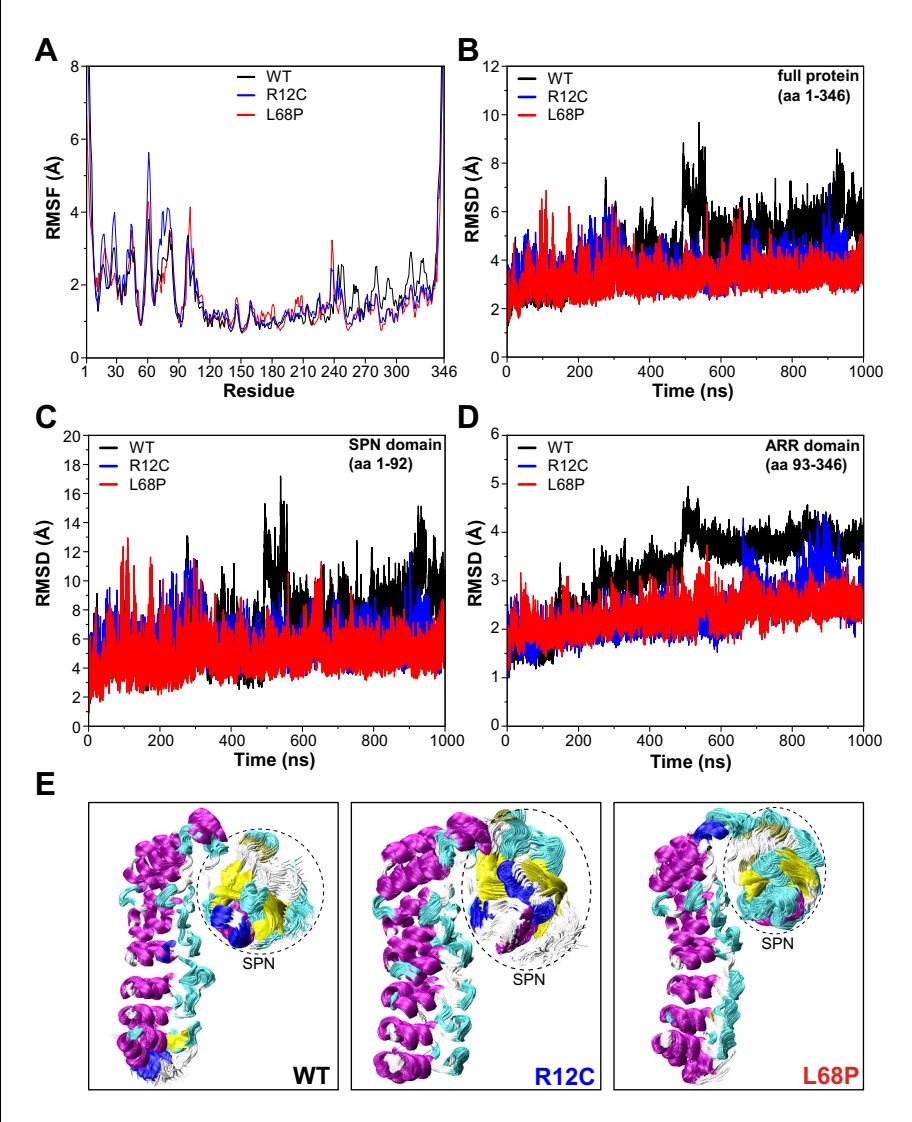

**Figure 4.** SHANK3 mutants show altered intra-domain conformational dynamics. (A) $C_\alpha$ backbone RMSF calculated for the trajectories of the SHANK3[(1-346)] fragment (PDB: 5G4X) either as WT or carrying one of the ASD-associated mutations R12C (blue) or L68P (red) show increased fluctuations for both mutants predominantly within the first 100 residues of the protein possibly suggesting a local mutation-induced increase in protein backbone dynamics. (B) $C_\alpha$ backbone RMSD determined for the full protein (aa 1–346). Within the first 100ns both mutants tend to show slightly increased backbone dynamics. This effect inverts with extended sampling time and mutants show on average reduced conformational dynamics compared to the WT. (C) $C_\alpha$ backbone RMSD restricted to the SPN domain of SHANK3 (aa 1–92). The traces resemble those observed for the full protein (aa 1–346) but absolute RMSD values are considerably larger for the SPN domain. (D) $C_\alpha$ backbone RMSD restricted to the ARR domain of SHANK3 (aa 93–346). The WT protein shows a stepwise increase in RMSD over time possibly indicating switches between distinct conformational states. This behavior is largely missing in both mutants but is more prominent for the L68P variant. (E) Local hotspots of conformational dynamics can be seen within the SPN domain (encircled) from overlays of individual frames of the trajectory (every fifth frame loaded, overlay [beginning:step:end] =0:20:20002 resulting in 1000 frames). RMSF = root mean square fluctuation, WT = wild type, ASD = autism spectrum disorders, RMSD = root mean square deviation.

The online version of this article includes the following video and source data for figure 4:

**Source data 1.** RMSF and RMSD analysis of MD trajectories.

**Figure 4—video 1.** MD trajectory of the SHANK3[(1-346)] WT fragment.

https://elifesciences.org/articles/66165#fig4video1

**Figure 4—video 2.** MD trajectory of the SHANK3[(1-346)] R12C fragment.

*Figure 4 continued on next page*

*Figure 4 continued*

https://elifesciences.org/articles/66165#fig4video2

**Figure 4—video 3.** MD trajectory of the SHANK3$^{(1-346)}$ L68P fragment.

https://elifesciences.org/articles/66165#fig4video3

independent of the genotype (*Figure 5E*). We thus asked whether these dendritic SHANK3 clusters are contacted by a presynaptic terminal by performing immunostaining for homer and bassoon (*Figure 5H and I*). Interestingly, we observed that approximately 48–54% of dendritic clusters of both SHANK3 mutants co-localize with bassoon (*Figure 5F*) and we found that approximately 32–34% of these dendritic SHANK3 clusters co-localize with homer (*Figure 5G*). Collectively, these data suggest that both mutations lead to an increased formation of dendritic clusters that are in part associated with shaft synapses.

## Synaptic turnover of SHANK3 and its interaction partner cortactin is affected by ASD-associated mutations of SHANK3

Mutation-induced alterations in SHANK3 structure and localization might also impact its ability to associate with synaptic binding partners. We therefore tested if SHANK3 mutants differentially interact with cortactin, homer1, or SAPAP1 (*Boeckers et al., 1999*; *Naisbitt et al., 1999*; *Tu et al., 1999*). To this end, we performed a set of co-immunoprecipitation (co-IP) experiments from HEK293 cells overexpressing both, a C-terminally RFP-tagged SHANK3$^{FL}$ variant (WT, R12C, or L68P) and one of the GFP-tagged binding partners. Interestingly, mutant SHANK3 variants showed no significant changes in binding to cortactin, homer, or SAPAP1 (*Figure 6A–C*). We next aimed to test, whether these mutations result in a perturbation of protein dynamics at the synapse. We employed fluorescence recovery after photobleaching (FRAP) of GFP-SHANK3$^{FL}$ variants (WT or mutant) or the interaction partners cortactin and homer1 in dendritic spines to address this question. Additionally, we tested the neuronal activity-dependence of GFP-SHANK3 turnover in dendritic spines by treatment with (+)-Bicuculline, which blocks GABA$_A$ receptor-mediated inhibitory currents and thus indirectly stimulates excitatory neurotransmission (*Nowak et al., 1982*). Strikingly, we found that the R12C, but not the L68P mutant, exhibited a strongly increased recovery, corresponding to a profound reduction in the synaptic residing time and thus increased turnover rate (*Figure 7A and D*). As expected, all tested GFP-SHANK3$^{FL}$ variants displayed a slower recovery upon bicuculline treatment, which might arise from activity-induced oligomerization of SHANK3 at the PSD (*Grabrucker, 2014*). The observed increase in synaptic turnover of R12C mutant is not a result of a mutation-induced enhancement of de novo protein synthesis as demonstrated by fluorescent noncanonical amino acid tagging (FUNCAT, *Figure 7—figure supplement 1*). When we analyzed the turnover of homer1 in the presence of ASD-associated SHANK3 mutants, we did not observe differences in the mobile fraction or recovery kinetics (*Figure 7B and E*). However, cortactin showed a significantly reduced mobile fraction in the presence of the L68P mutant (*Figure 7C and F*).

Additionally, the shape of the FRAP curve indicated a change of cortactin recovery kinetics in the presence of either mutant. Taken together, the FRAP experiments indicate altered diffusion dynamics of the SHANK3 R12C mutant in dendritic spines, corresponding to a reduced synaptic residing time. Also, cortactin shows faster recovery kinetics in the presence of the R12C mutant. Although the turnover of the SHANK3 L68P mutant in spines does not differ from the WT, the presence of the L68P mutant induces a reduction of cortactin recovery.

## Discussion

### ASD-associated SHANK3 missense variants exhibit distinguishable structural perturbations on secondary, tertiary, and quaternary structure level

The precise regulation of synaptic structure and function is crucial to maintain neuronal circuit integrity. Perturbation or dysregulation of synaptic proteins constitutes an integral part of many neurodevelopmental and neuropsychiatric diseases such as ASD (*Durand et al., 2007*; *Kleijer et al., 2014*). Although attempts have been made to converge existing data to a few common ASD pathways, the

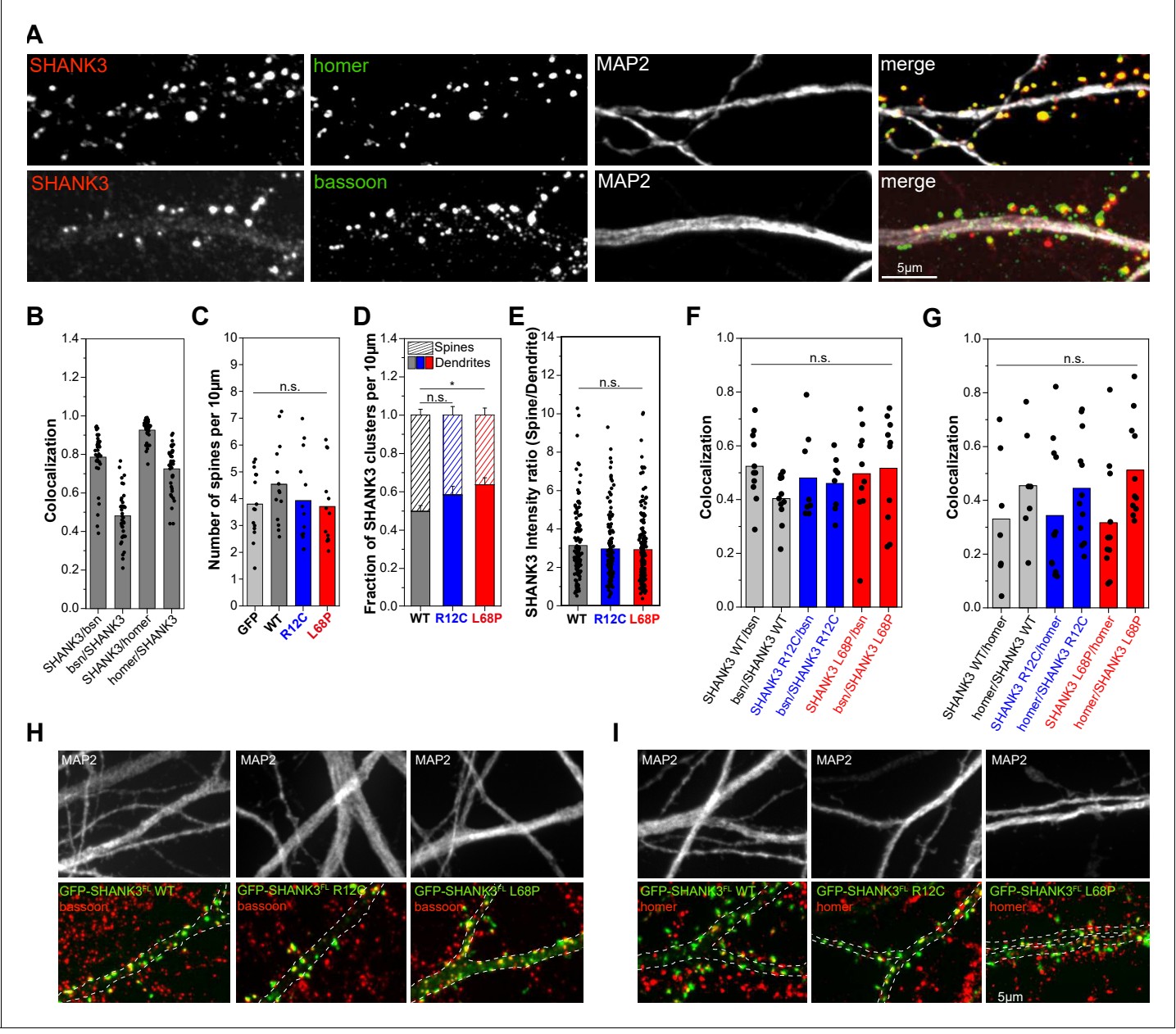

**Figure 5.** Mutant SHANK3 variants form more clusters in dendritic shaft. (**A**) Representative images of immunostained endogenous SHANK3 (red), homer or bassoon (green), and MAP2 (gray). (**B**) Quantification of the co-localization (SHANK3/bsn and bsn/SHANK3: n = 36 cells from two independent cultures; SHANK3/homer and homer/SHANK3: n = 36 cells from two independent cultures). Bars are showing the mean as well as individual data points. Approximately 72% of homer-positive spines are also SHANK3-positive, while ~93% of SHANK3-positive spines co-localize with homer. Additionally, ~79% of SHANK3-positive spines co-localize with the presynaptic marker bassoon. (**C**) Quantification of the total number of spines per 10 µm in primary rat hippocampal neurons overexpressing GFP-SHANK3$^{FL}$ variants for 16–18 hr (GFP: 15 cells from four independent cultures, WT: 14 cells from three independent cultures, R12C: 13 cells from four independent cultures, L68P: 13 cells from four independent cultures). Bars are showing the mean as well as individual data points. No significant differences in average spine numbers is found (Kruskal-Wallis ANOVA with Dunn's post hoc test: p=0.59–1). (**D**) Analysis of SHANK3 cluster distribution in neurons overexpressing GFP-SHANK3$^{FL}$ variants (same dataset as in (**C**)). Bars are showing mean ± SEM. No significant effect of the R12C mutation on the relative distribution of SHANK3 clusters between spines and dendrites is found. The L68P mutation caused a significant increase in dendritic SHANK3 clusters (Kruskal-Wallis ANOVA with Dunn's post hoc test: p(WT/R12C) =0.42; p(WT/L68P)=0.04 and p(R12C/L68P)=0.93 at p=95%). (**E**) Line profile analysis of the SHANK3 distribution between spines and dendrites (WT: 12 cells from four independent cultures, R12C: 11 cells from four independent cultures, L68P: 12 cells from five independent cultures). Bars are showing the mean as well as individual data points. No differences in the mean SHANK3 intensity ratio between spines and dendrites are found (one-way ANOVA with Bonferroni multiple comparisons test: p(WT/R12C)=1 and p(WT/L68P)=0.83 at p=95%). (**F**) Quantification of bassoon (bsn)/SHANK3 and SHANK3/ bsn co-localization on dendritic clusters of neurons overexpressing GFP-SHANK3$^{FL}$ variants (WT: n = 11 cells from three independent cultures; R12C:

*Figure 5 continued on next page*

*Figure 5 continued*

n = 8 cells from three independent cultures; L68P: n = 10 cells from three independent cultures). No significant differences between genotypes are found (Kruskal-Wallis ANOVA with Dunn's post hoc test: SHANK3/bsn co-localization: p(WT/R12C)=1; p(WT/L68P)=1; and p(R12C/L68P)=1 at p=95%; bsn/SHANK3 co-localization: p(WT/R12C)=1; p(WT/L68P)=1; and p(R12C/L68P)=1 at p=95%). Bars are showing the mean as well as individual data points per cell. (G) Quantification of homer/SHANK3 and SHANK3/homer co-localization on dendritic clusters of neurons overexpressing GFP-SHANK3$^{FL}$ variants (WT: n = 7 cells from two independent cultures; R12C: n = 12 cells from two independent cultures; L68P: n = 11 cells from two independent cultures). No significant differences between genotypes are found (Kruskal-Wallis ANOVA with Dunn's post hoc test: SHANK3/homer co-localization: p(WT/R12C)=1; p(WT/L68P)=1; and p(R12C/L68P)=1 at p=95%; homer/SHANK3 co-localization: p(WT/R12C)=1; p(WT/L68P)=1; and p (R12C/L68P)=1 at p=95%). Bars are showing the mean as well as individual data points. (H) Representative images of GFP-SHANK3$^{FL}$ expressing neurons, co-stained for MAP2 and bassoon. The MAP2 signal was used as a mask to draw the outline of dendrites. Co-localization of GFP-SHANK3$^{FL}$ clusters with bassoon was analyzed within the MAP2 mask. (I) Representative images of GFP-SHANK3$^{FL}$ expressing neurons, co-stained for MAP2 and homer. Co-localization of GFP-SHANK3$^{FL}$ clusters with homer was analyzed within the MAP2 mask. WT = wild type.

The online version of this article includes the following source data and figure supplement(s) for figure 5:

**Source data 1.** Colocalization of endogenous SHANK3 with bassoon and homer.

**Source data 2.** Quantification of SHANK3 clusters and spine numbers.

**Source data 3.** SHANK3 intensity ratio (spine vs. dendrite).

**Source data 4.** Colocalization of overexpressed SHANK3 variants with bassoon and homer.

**Figure supplement 1.** Representative images of hippocampal primary neurons overexpressing GFP-SHANK3$^{FL}$ variants.

complexity of these pathways remains a challenge and is far from being fully understood (*Luo et al., 2018*). Due to the high frequency of mutations in the synaptic scaffolding protein SHANK3 (more than 1 in 50) in patients with ASD and intellectual disability, mutation screening of SHANK3 has been suggested for consideration in clinical practice (*Leblond et al., 2014*). Importantly, it has been emphasized previously that more than one pathogenic pathway downstream of SHANK3 is likely to exist and individual pathogenic processes related to synaptic organization and function have been described elsewhere (*Durand et al., 2012*; *Mameza et al., 2013*; *Qin et al., 2018*; *Wang et al., 2020a*). Therefore, it seems likely that individual ASD-associated pathogenic pathways related to SHANK3 are reflected by distinguishable structural perturbations elicited by distinct missense mutations. Hence, a precise understanding of SHANK3 missense mutation-induced structural changes as well as their associated pathogenic pathways should reveal disease-causing mechanisms but is largely missing. Bioinformatics studies suggested that functional consequences of missense mutations can be predicted based on structural features for the ASD- and cancer-associated protein phosphatase PTEN, as well as for voltage-gated sodium and calcium channels (*Heyne et al., 2020*; *Smith et al., 2019a*; *Smith et al., 2019b*). However, to our knowledge, studies correlating distinguishable structural perturbations of disease-relevant protein missense variants with an altered cellular phenotype are very limited (*Post et al., 2020*). In this work, we found that missense mutation-induced impairments of the structural integrity of SHANK3 serve as molecular starting point for higher-order pathogenic processes associated with ASD. Taken together, we demonstrate changes in synaptic turnover as exemplary consequence of ASD-associated missense mutations, indicating that structural impairments directly translate to cellular alterations.

## R12C and L68P mutations result in altered conformational flexibility of SHANK3 on a nanosecond timescale

The description of the functional impact of protein missense mutation is incomplete if only a static structural picture is considered. Consequently, the evaluation of the pathogenic role of protein missense variants greatly benefits from analyses of changes in structural dynamics as a result to these point mutations (*Ponzoni and Bahar, 2018*). Here, we identified the SPN domain as a local hotspot of conformational dynamics and found that both ASD-associated missense mutations further increase conformational fluctuations within the SPN domain. Furthermore, we demonstrated a reduction of conformational dynamics within the ARR domain of SHANK3 due to the R12C and L68P mutation, which suggests the existence of more wide-ranging structural interactions affecting distal domains. Consistently, reconstruction of the L68P mutant His$_6$-SUMO-SHANK3$^{(1-676)}$ complex topology from solution scattering data revealed a rearrangement of ARR domains compared to the WT complex (*Figure 1F*; *Figure 1—figure supplement 4*). Thus, local and distal structural disturbances in response to ASD-associated SHANK3 missense mutations were observed, which could explain

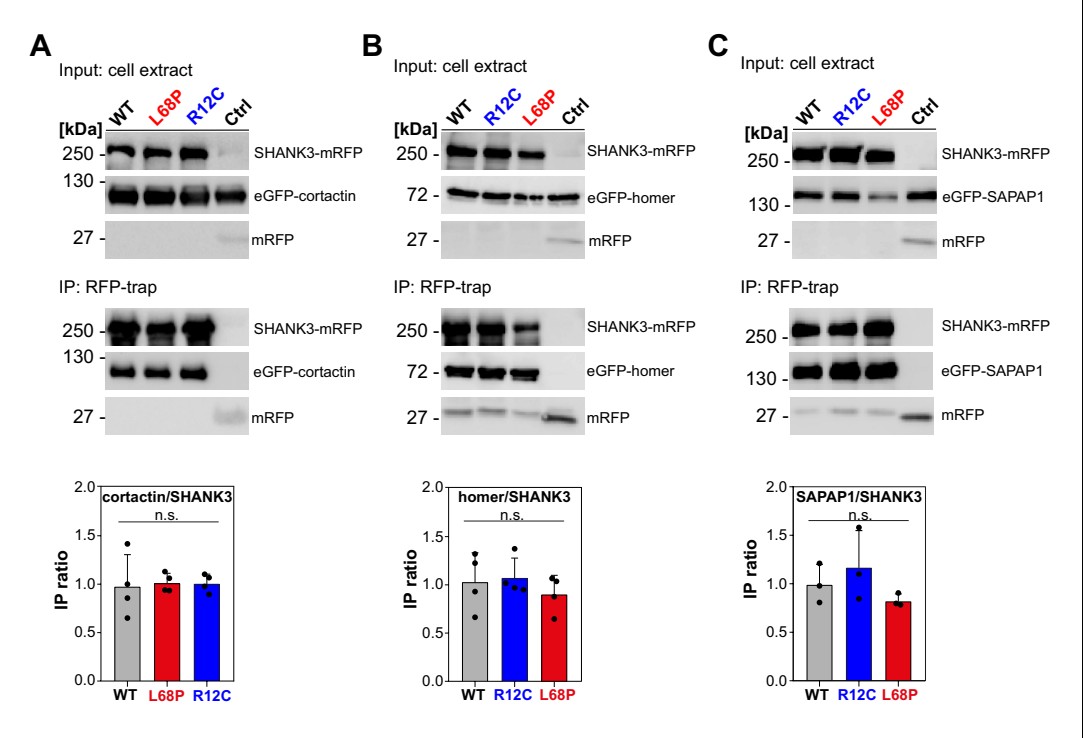

**Figure 6.** Mutant SHANK3 variants readily interact with cortactin, homer, and SAPAP1. (**A**) A representative western blot from co-IP assays of SHANK3 and cortactin from HEK293 cells is shown (detection SHANK3: rat-α-mRFP, 1 s; detection cortactin: mouse-α-GFP, 1 s; n = 4 independent experiments). No difference in binding of cortactin to ASD-associated SHANK3 mutants can be detected compared to the WT (one-way ANOVA with Bonferroni multiple comparisons test: p(WT/R12C)=1, p(WT/L68P)=1, and p(R12C/L68P)=1 at p=95%). (**B**) Representative western blot from co-IPs of SHANK3 and homer (detection SHANK3: rat-α-mRFP, 1 s; detection homer: mouse-α-GFP, 1.5 s; n = 4 independent experiments). No difference in binding of homer to the tested SHANK3 mutants can be found compared to the WT (one-way ANOVA with Bonferroni multiple comparisons test: p(WT/R12C)=1, p(WT/L68P)=1, and p(R12C/L68P)=0.99 at p=95%). (**C**) Representative western blot from co-IPs of SHANK3 and SAPAP1 (detection SHANK3: rat-α-mRFP, 9 s; detection SAPAP1: mouse-α-GFP, 2 s; n = 3 independent experiments). No difference in binding of SAPAP1 to the SHANK3 mutants can be detected compared to the WT (one-way ANOVA with Bonferroni multiple comparisons test: p(WT/R12C)=1, p(WT/L68P)=1, and p(R12C/L68P)=0.41 at p=95%). co-IP = co-immunoprecipitation, WT = wild type, ASD = autism spectrum disorders.

The online version of this article includes the following source data for figure 6:

**Source data 1.** Quantification of western blot data.

changes in protein function not directly related to the immediate locus of amino acid substitution. Therefore, point mutations could impact overall 3D protein structure resulting in altered intra- and intermolecular interactions.

## SHANK3 missense mutants exhibit altered localization and residing time at the synapse

As SHANK3 is critically involved in regulating synaptic structure and function, neurons are sensitive to altered gene dosage, which has been pointed out by several studies (*Bozdagi et al., 2010*; *Durand et al., 2007*; *Wang et al., 2020b*; *Yi et al., 2016*). Of note, albeit both mutations are located within the SPN domain, they had distinct effects on SHANK3 folding and residing time as well as localization at the synapse, suggesting that each point mutation might cause synaptic dysfunction by different mechanisms. Since autistic patients carrying the SHANK3 missense mutations studied here are heterozygous, the question arises whether these mutations render the protein dysfunctional and thereby reduce the minimum number of functional proteins that is necessary for normal synaptic transmission. A profound decrease in synaptic residing time of R12C SHANK3 mutant is

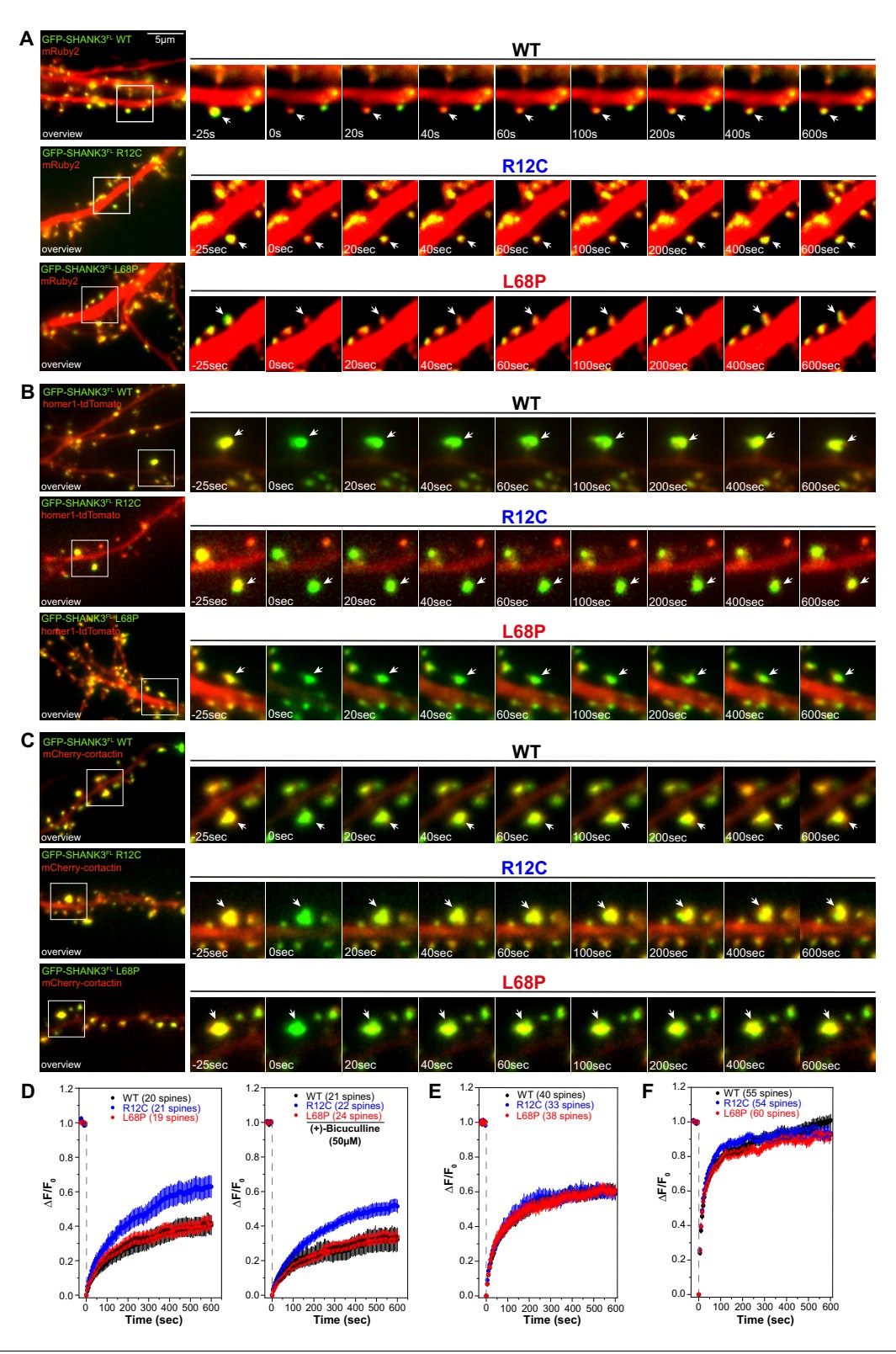

**Figure 7.** Mutations of SHANK3 differentially affect the protein residing time in spines and dynamics of distinct interaction partners. (**A**) Representative images of photobleached spines (488 nm) are shown at selected time points to illustrate the dynamics of GFP-SHANK3^FL variants in primary rat hippocampal neurons. (**B**) Representative images of photobleached spines (561 nm) co-expressing homer1-tdTomato and GFP-SHANK3^FL (WT, R12C, or L68P) are shown at selected timepoints during FRAP to illustrate the dynamics of homer1 in the presence of different SHANK3 variants. (**C**)

*Figure 7 continued on next page*

*Figure 7 continued*

Representative images of photobleached spines (561 nm) co-expressing mCherry-cortactin and GFP-SHANK3$^{FL}$ (WT, R12C, or L68P) are shown at selected timepoints during FRAP to illustrate the dynamics of cortactin in the presence of different SHANK3 variants. (D) FRAP curves demonstrating the dynamics of mutant SHANK3 in spines. The R12C mutant is displaying an increased recovery corresponding to a reduced residing time in spines (Kruskal-Wallis ANOVA with Dunn's post hoc test; untreated: p(WT/R12C)<0.0001, p(R12C/L68P)<0.0001, and p(WT/L68P)=1; bic: p(WT/R12C)<0.0001, p(R12C/L68P)<0.0001, and p(WT/L68P)=1). All tested SHANK3 variants show a mildly decreased recovery after pharmacological stimulation with (+)-Bicucullin (5 min, 37°C, 5% $CO_2$; data collected from three independent cultures for control curves and two independent cultures for (+)-Bicucullin stimulation). (E) FRAP curves showing the dynamics of homer1 in the presence of different GFP-SHANK3$^{FL}$ variants (WT, R12C, L68P). ASD-associated SHANK3 mutants cause no difference in the recovery of homer1 (Kruskal-Wallis ANOVA with Dunn's post hoc test; p(WT/R12C)=1, p(R12C/L68P)=1, and p(WT/L68P)=1). (F) FRAP curves showing the dynamics of cortactin in the presence of different GFP-SHANK3$^{FL}$ variants (WT, R12C, L68P). Qualitatively, both SHANK3 mutants alter the recovery kinetics of cortactin. Additionally, the L68P mutant significantly reduces the mobile fraction of cortactin in spines (Kruskal-Wallis ANOVA with Dunn's post hoc test; p(WT/R12C)=1, p(R12C/L68P)=0.0002, and p(WT/L68P)=0.0007). WT = wild type, FRAP = fluorescence recovery after photobleaching.

The online version of this article includes the following source data and figure supplement(s) for figure 7:

**Source data 1.** FRAP of overexpressed SHANK3 variants under basal conditions.
**Source data 2.** FRAP of overexpressed SHANK3 variants upon stimulation with (+)-Bicuculline.
**Source data 3.** FRAP of overexpressed homer1.
**Source data 4.** FRAP of overexpressed cortactin.
**Figure supplement 1.** Global de novo protein synthesis in hippocampal primary neurons overexpressing GFP-Shank3$^{FL}$ variants monitored by fluorescence non-canonical amino acid tagging (FUNCAT).

likely to destabilize the synaptic SHANK3 pool and could thus elicit a haploinsufficiency-like phenotype.

Moreover, an increased synaptic turnover of the R12C mutant might also alter the localization and dynamics of some SHANK3 binding partners which can result in altered synaptic function. Interestingly, we observed that both mutants changed the recovery kinetics of cortactin, which is known to link dynamics of the PSD to rearrangements of the F-actin cytoskeleton via interaction with SHANK3 (*MacGillavry et al., 2016*). Cortactin can stabilize F-actin and promote the formation of branched actin filaments, thereby directly influencing plasticity and stability of dendritic spines. Temporal dynamics and the cortactin concentration at the synapse are highly regulated (*Mikhaylova et al., 2018*). Thus, SHANK3-induced deviations in cortactin turnover are likely to have consequences for dendritic spine organization and functions.

Furthermore, we found an increase in the number of dendritic SHANK3 clusters for the L68P mutation. About a half of these clusters represent synapses as they were opposed by a presynaptic terminal. Notably, only approximately one third of SHANK3-positive dendritic shaft clusters co-localize with homer independent of the tested SHANK3 variant, while this co-localization is close to 100% in dendritic spines. However, we observed no differences in mutant SHANK3-homer1 interactions nor detected any SHANK3 mutation-induced changes in the mobile fraction of homer1 in FRAP experiments. This suggests that other factors might contribute to the differential recruitment of homer1 to shaft and spine synapses.

In this scenario, structurally perturbed or destabilized proteins which are not removed by cellular protein quality control mechanisms will be incorporated into the PSD and result in changes of synaptic proteostasis, protein-protein interactions and signaling (*Reim et al., 2017*). Previous studies indicated that both gain- and loss-of-function with respect to interaction partners are likely to contribute to the pathogenicity of SHANK3 missense mutations. The L68P mutation for instance has been demonstrated to result in increased binding to ARR domain ligands such as sharpin or α-fodrin, whereas both the R12C and L68P mutation reduce binding to SPN domain ligands including the Ras superfamily (*Lilja et al., 2017*; *Mameza et al., 2013*). Unfortunately, the majority of *SHANK3*-associated mouse models used for the study of ASD are based on a full or partial knockout that will not resemble the structure-function relationships of the disease-causing mutations (*Lee et al., 2015*). Accordingly, a corresponding R12C or L68P mutant knock-in mouse line would be of great value to study such scenarios of ASD-associated pathogenic mechanisms in the future.

# Materials and methods

**Key resources table**

| Reagent type (species) or resource | Designation | Source or reference | Identifiers | Additional information |
|---|---|---|---|---|
| Antibody | Guinea pig polyclonal Shank3 | Synaptic Systems | 162304 | for ICC (1:500) |
| Antibody | Rabbit polyclonal Homer-1,2,3 | Synaptic Systems | 160103 | for ICC (1:500) |
| Antibody | Chicken polyclonal Bassoon | Synaptic Systems | 141016 | for ICC (1:500) |
| Antibody | Mouse monoclonal MAP2-AF488 | Merck Millipore | MAB3418X | for ICC (1:500) |
| Antibody | Mouse monoclonal GFP | Covance | MMS-118P-500 | for Western Blot (1:3000) |
| Antibody | Rat monoclonal RFP | Chromotek | 5F8 | for Western Blot (1:1000) |
| Antibody | Rabbit polyclonal IgG anti-chicken IgY (H + L)-Cy3 | Dianova | 303-165-003 | for ICC (1:500) |
| Antibody | Goat polyclonal anti-rabbit IgG (H + L)-AF568 | Life Technologies | A-11036 | for ICC (1:500) |
| Antibody | Goat polyclonal anti-guinea pig IgG-STAR 635P | Abberior | ST635P-1006 | for ICC (1:500) |
| Antibody | Goat polyclonal anti-rat IgG (H + L)-HRP | Jackson ImmunoResearch | 112-035-062; RRID:AB_2338133 | for Western Blot (1:2500) |
| Antibody | Goat polyclonal anti-mouse IgG (H + L)-HRP | Jackson ImmunoResearch | 115-035-003; RRID:AB_10015289 | for Western Blot (1:2500) |
| Strain, strain background (*Escherichia coli*) | XL10 gold | Stratagene | 200314 | for plasmid DNA production; stock amplified in the lab |
| Strain, strain background (*Escherichia coli*) | BL21(DE3) | ThermoFisher Scientific | K30001 | for protein purification; stock amplified in the lab |
| Strain, strain background (*Escherichia coli*) | Stbl2 | Invitrogen | 10268–019 | obtained from Matthias Kneussel; stock amplified in the lab |
| Cell line (*Homo-sapiens*) | Embryonic kidney (HEK 293T) | ATCC | CRL-3216; RRID:CVCL_0063 | for co-IP |
| Transfected construct (rat) | pET-His$_6$-SUMO-SHANK3$^{(1-676)}$ WT | This paper | | for bacterial protein production; available from Hans-Jürgen Kreienkamp |
| Transfected construct (rat) | pET-His$_6$-SUMO-SHANK3$^{(1-676)}$ R12C | This paper | | for bacterial protein production; available from Hans-Jürgen Kreienkamp |
| Transfected construct (rat) | pET-His$_6$-SUMO-SHANK3$^{(1-676)}$ L68P | This paper | | for bacterial protein production; available from Hans-Jürgen Kreienkamp |
| Transfected construct (rat) | pHAGE2-EF1a-EGFP-SHANK3$^{FL}$ WT | This paper | | for transfection of primary hippocampal neurons; available from Hans-Jürgen Kreienkamp |
| Transfected construct (rat) | pHAGE2-EF1a-EGFP-SHANK3$^{FL}$ R12C | This paper | | for transfection of primary hippocampal neurons; available from Hans-Jürgen Kreienkamp |

*Continued on next page*

*Continued*

| Reagent type (species) or resource | Designation | Source or reference | Identifiers | Additional information |
|---|---|---|---|---|
| Transfected construct (rat) | pHAGE2-EF1a-EGFP-SHANK3$^{FL}$ L68P | This paper | | for transfection of primary hippocampal neurons; available from Hans-Jürgen Kreienkamp |
| Transfected construct (rat) | pmRFP-N3-SHANK3$^{FL}$ WT | doi:10.1038/ncb3487 | | for transfection of HEK293 cells and co-IP |
| Transfected construct (rat) | pmRFP-N3-SHANK3$^{FL}$ R12C | doi:10.1038/ncb3487 | | for transfection of HEK293 cells and co-IP |
| Transfected construct (rat) | pmRFP-N3-SHANK3$^{FL}$ L68P | doi:10.1038/ncb3487 | | for transfection of HEK293 cells and co-IP |
| Transfected construct (rat) | pEGFP-C1-Homer1b | This paper | | for transfection of HEK293 cells and co-IP; available from Hans-Jürgen Kreienkamp |
| Transfected construct (rat) | pEGFP-C1-SAPAP1 | This paper (cDNA coding for rat SAPAP1 obtained from Stefan Kindler) | | for transfection of HEK293 cells and co-IP; available from Hans-Jürgen Kreienkamp |
| Transfected construct (human) | pEGFP-C1-cortactin | This paper | | for transfection of HEK293 cells and co-IP; available from Hans-Jürgen Kreienkamp |
| Transfected construct (mouse) | mCherry-cortactin-C-12 | doi:10.1016/j.neuron.2018.01.046 | Addgene plasmid #55020 | for transfection of primary hippocampal neurons |
| Transfected construct (rat) | pFUGW-f(syn)Homer1-tdTomato-WPRE | A gift from Christian Rosenmund | BL-1034 | for transfection of primary hippocampal neurons |
| Transfected construct | pAAV-Syn-mRuby2 | doi:10.15252/embj.2018101183 | | for transfection of primary hippocampal neurons |
| Transfected construct | pMAX-GFP | Lonza Bioscience | V4XP-3012 | for transfection of primary hippocampal neurons |
| Transfected construct (human) | pETM11-T7-His$_6$-SenP2 | EMBL Heidelberg | | for recombinant protein tag removal |
| Peptide, recombinant protein | His$_6$-SUMO-SHANK3$^{(1-676)}$ WT | This paper | | purified from BL21(DE3) using pET-His$_6$-SUMO-SHANK3$^{(1-676)}$ WT |
| Peptide, recombinant protein | His$_6$-SUMO-SHANK3$^{(1-676)}$ R12C | This paper | | purified from BL21(DE3) using pET-His$_6$-SUMO-SHANK3$^{(1-676)}$ R12C |
| Peptide, recombinant protein | His$_6$-SUMO-SHANK3$^{(1-676)}$ L68P | This paper | | purified from BL21(DE3) using pET-His$_6$-SUMO-SHANK3$^{(1-676)}$ L68P |
| Peptide, recombinant protein | SHANK3$^{(1-676)}$ WT | This paper | | purified from BL21(DE3), see protein purification methods |
| Peptide, recombinant protein | SHANK3$^{(1-676)}$ R12C | This paper | | purified from BL21(DE3), see protein purification methods |
| Peptide, recombinant protein | SHANK3$^{(1-676)}$ L68P | This paper | | purified from BL21(DE3), see protein purification methods |
| Peptide, recombinant protein | His$_6$-SenP2 | This paper | | purified from BL21(DE3); for recombinant protein tag removal; see protein purification methods |
| Chemical compound, drug | Profinity IMAC resin, Ni$^{2+}$-charged | Bio-Rad | 1560135 | for protein purification |

*Continued on next page*

*Continued*

| Reagent type (species) or resource | Designation | Source or reference | Identifiers | Additional information |
|---|---|---|---|---|
| Chemical compound, drug | RFP-trap agarose | Chromotek | rta-20 | for co-IP |
| Chemical compound, drug | (+)-Bicuculline | Tocris Bioscience | 0130 | for treatment of hippocampal primary neurons |
| Chemical compound, drug | Tetrodotoxin | Tocris Bioscience | 1078 | for treatment of hippocampal primary neurons |
| Chemical compound, drug | D-APV | Tocris Bioscience | 0106 | for treatment of hippocampal primary neurons |
| Chemical compound, drug | Lipofectamine 2000 | ThermoFisher Scientific | 11668019 | for transfection of primary hippocampal neurons |
| Chemical compound, drug | TurboFect transfection reagent | ThermoFisher Scientific | R0532 | for transfection of HEK293 cells |
| Commercial assay or kit | BCA protein assay | ThermoFisher Scientific | 23250 | for determination of protein concentration |
| Commercial assay or kit | Plasmid midi kit | Qiagen | 12145 | for purification of plasmid DNA |
| Commercial assay or kit | WesternBright chemiluminescence substrate | Biozym | 541003 | for western blot detection |
| Software, algorithm | VMD 1.9.3 | doi:10.1016/0263-7855(96)00018-5 | | for analysis and visualization of molecular graphics |
| Software, algorithm | Origin 2019b | OriginLab | | for graphing and data analysis |
| Software, algorithm | PyMOL 2.3.4 | Schrödinger, Inc. | | for molecular visualization |
| Software, algorithm | ImageJ/Fiji | NIH | | for image processing and analysis |
| Software, algorithm | Adobe Illustrator 2021 | Adobe Inc. | | for graphing and visualization |
| Software, algorithm | ATSAS 3.0.1 | doi:10.1107/S1600576720013412 | | for analysis of SAXS data |
| Software, algorithm | SnapGene | GSL Biotech LLC | | for plasmid cloning |
| Software, algorithm | Image Lab | Bio-Rad | | for processing and analysis of western blot images |
| Software, algorithm | POV-Ray 3.7 | Other | | for ray tracing of molecular graphics |
| Software, algorithm | MATLAB | MathWorks | | for SHANK3 intensity distribution analysis (*Figure 5E*) |

Antibodies, expression constructs, bacterial strains, recombinant proteins, and other reagents are listed in the key resource table.

## Protein expression and purification

For bacterial expression and purification of His$_6$-SUMO-tagged SHANK3$^{(1-676)}$ variants, chemically competent *E. coli BL21(DE3)* cells were transformed with the corresponding pET vectors. Recombinant protein expression was induced at $OD_{600}$ ~0.5 with 300 µM isopropyl-β-D-thiogalactopyranoside (IPTG) in the presence of kanamycin and was continued for 16–18 hr at 18˚C. Importantly, all subsequent steps were performed either on ice or at 4˚C to enhance protein stability. Bacteria were harvested and resuspended in the following buffer: 250 mM NaH$_2$PO$_4$, 250 mM NaCl, 0.5 mM DTT, 10 mM Imidazole, 1x Complete Protease Inhibitor Cocktail, pH = 6.5. Subsequently, bacterial lysis was performed by treatment with lysozyme followed by mild sonication, a freeze-thaw cycle and another sonication step. His$_6$-tagged proteins were then captured from the lysate by Ni$^{2+}$-IDA

chromatography (Profinity IMAC resin, Bio-Rad) under gravity flow and washed with 35 mM Imidazole. For CD and nDSF measurements, protease inhibitors were removed prior to elution by additional washes with inhibitor-free washing buffer to facilitate SUMO-protease activity for recombinant tag removal. Batch elution was performed with 200 mM Imidazole. To this preparation the home-made SUMO protease His$_6$-SenP2 was added to a final concentration of approximately 0.3 mg/mL and proteins were incubated for 2 hr at 4°C under gentle rotation for cleavage of the His$_6$-SUMO tag. Following addition of Complete Protease Inhibitor Cocktail to inhibit any further proteolytic activity, the preparation was diluted to a final Imidazole concentration of approximately 30 mM and was loaded twice onto a freshly packed Ni$^{2+}$-IDA column to remove both His$_6$-SenP2 as well as the cleaved His$_6$-SUMO tag. We noted that SHANK3$^{(1-676)}$ variants tend to aggregate and precipitate more easily at higher concentrations after removal of SUMO, possibly due to reduced protein solubility. We therefore kept protein concentrations lower compared to preparations where the tag has not been removed. Subsequently, dilute proteins were concentrated using Amicon Ultra 15 mL centrifugal filters (Merck Millipore, 10 kDa NMWCO) and subjected to single-step buffer exchange into SEC running buffer (see 'SAXS method') over a PD10 desalting column (GE Healthcare, Sephadex G-25 M matrix) under gravity flow. Eluted proteins were finally up-concentrated and stored on wet ice.

## Size exclusion chromatography and small-angle X-ray scattering

For SAXS, bacterially expressed His$_6$-SUMO-SHANK3$^{(1-676)}$ variants were pre-purified over Ni$^{2+}$-IDA under gravity flow and eluted in 500–750 µL fractions with 250 mM NaH$_2$PO$_4$ (pH = 6.5), 250 mM NaCl, 0.5 mM DTT, 200 mM Imidazole, and 1x complete protease inhibitor cocktail (Merck Millipore). Subsequently, proteins were subjected to SEC at 4°C with a flow rate of 0.4 mL/min (running buffer 100 mM NaH$_2$PO$_4$, 100 mM NaCl, 0.5 mM DTT, pH = 6.5) using a Superdex 75 10/300 GL column (GE Healthcare). To establish a protein dilution series, individual SEC peak fractions were pooled and pre-concentrated with an Amicon Ultra 4 mL centrifugal filter (Merck Millipore, 30 kDa NMWCO). Subsequently, protein concentration was determined spectrophotometrically based on 280 nm absorption using SEC running buffer for blank subtraction. Subsequently, samples were accordingly diluted and measured in SEC running buffer.

SAXS measurements (I(s) vs. s with s[nm$^{-1}$]=4πsin(θ)/λ and 2θ being the scattering angle) were performed at the EMBL-P12 bioSAXS beam line (PETRAIII, DESY, Hamburg) (*Blanchet et al., 2015*) under default operating conditions (10 keV, λ = 0.124 nm, Pilatus 2M detector with a distance of 3.0 m) at 10°C. Data was collected in the s-range of 0.0287–7.267 for 20 successive frames with an exposure time of 0.5 s per frame. Primary data processing included radial averaging, normalization, and buffer subtraction. Subsequently, data analysis was performed with the software package ATSAS 3.0.1 (*Franke et al., 2017*).

For experimental SAXS profiles, the radius of gyration was calculated using the Guinier approximation with automated detection of a suitable Guinier range by *AUTORG* (*Petoukhov et al., 2007*). The R$_g$ values of the WT and R12C mutant variant of SHANK3 showed a linear concentration dependence so that subsequent analyses were performed on SAXS profiles, which were extrapolated to zero concentration. The real space pair-distance distribution functions (PDDFs, P(r) vs. r profiles) were approximated from the experimental scattering data via an indirect Fourier transform (IFT) procedure using the program *GNOM* (*Svergun, 1992*). Accordingly, the values for R$_g$ and D$_{max}$ were determined by using automated criteria for the Lagrange parameter and D$_{max}$. To model the topology of His$_6$-SUMO-SHANK3$^{(1-676)}$ variants from crystallographic structures of individual domains, the program *CORAL* was used (*Petoukhov et al., 2012*). Thereby, the employed crystallographic structures (SMT3(aa 13–98): chain C from PDB 2EKE, SPN-ARR: PDB 5G4X, SH3: PDB 5O99, PDZ: PDB 5OVA) are rotated and translated with certain restrictions to minimize the discrepancy between the experimental and the computed (model-derived) SAXS profile (*Duda et al., 2007*; *Lilja et al., 2017*; *Ponna et al., 2017*; *Ponna et al., 2018*). Dimeric subunit structures were split into corresponding monomers before using them as input for *CORAL*. For zero-extrapolated SAXS profiles, no symmetry restrictions were used (P1 symmetry) to model the topology of monomeric His$_6$-SUMO-SHANK3$^{(1-676)}$ variants, while P2 symmetry restrictions were employed for corresponding dimer models from SAXS data merged between highest and lowest concentrations. The atomic model of the SPN-ARR fragment was allowed to move freely while all other subunits were fixed. N- and C-termini as well as missing inter-domain linker regions were added to the crystallographic structures as

dummy residues (DRs) based on the ORF of His$_6$-SUMO-SHANK3$^{(1-676)}$ in the corresponding pET vector. To account for a potential mutation-induced disruption of the SPN/ARR domain interface, monomeric *CORAL* models were additionally created using a split SPN-ARR fragment, where the unstructured linker region (KRRVYAQNLI) has been replaced by the same number of DRs to allow flexible movement of both domains. The resulting structural models were visualized using PyMOL. SAXS data was deposited to SASBDB (WT: SASDLJ3, R12C: SASDLL3, and L68P: SASDLK3).

## Circular dichroism spectroscopy

Circular dichroism spectroscopy of SHANK3$^{(1-676)}$ variants was performed on a Chirascan circular dichroism spectrometer (Applied Photophysics; *Kelly et al., 2005*). Before measurement of CD spectra, pre-concentrated proteins in SEC running buffer (approximately 1 mg/mL) were buffer exchanged into 10 mM KH$_2$PO$_4$, 100 mM KF, 0.5 mM DTT, pH = 6.5 using PD10 desalting columns under gravity flow. Protein concentration was adjusted to 2 μM and CD spectra were recorded in the wavelength range of 175–260 nm at 10°C in a 1 mm quartz cuvette (0.5 nm step size, 3 s/point, three repeats). Due to low signal-to-noise ratio (SNR) <185 nm, far-UV spectra were restricted to the spectral range between 260 and 185 nm (*Figure 3—figure supplement 2*). Subsequent data processing was done as described elsewhere (*Greenfield, 2007c*).

Equilibrium chemical unfolding was done as described elsewhere (*Greenfield, 2007a*) with few modifications. Briefly, concentrated protein stock solutions as described above were diluted to approximately 2 μM and mixed with 10 M urea in SEC running buffer to establish a urea concentration series for each SHANK3$^{(1-676)}$ variant. Samples including blank were incubated on ice for at least 30 min and ellipticity was subsequently measured at 222 nm and at 10°C (3 s/point, 25 μs sample period, three repeats). After each CD measurement, the sample was recovered for subsequent assessment of protein concentration by BCA assay. The data was averaged, corrected for buffer absorption, and converted to the mean residue ellipticity [θ] as described elsewhere (*Greenfield, 2007c*). Subsequently, equilibrium chemical unfolding data was fitted to a two-state unfolding transition model given by the following equation (*Agashe and Udgaonkar, 1995*):

$$y_{obs} = \frac{y_N^0 + m_N[D] - y_U^0 - m_U[D]}{1 + e^{-\frac{\Delta G^0 + m_G[D]}{RT}}} + y_U^0 + m_U[D]$$

The parameters $y_N^0$, $y_U^0$ (intercepts), as well as $m_N$ and $m_U$ (slopes) were obtained separately by linear extrapolation of the corresponding native (N) and unfolded (U) baseline regions and were kept constant in the actual fitting procedure. Variable parameters obtained from the two-state unfolding transition fit were $\Delta G^0$ as well as the cooperativity parameter $m_G$. The free energy of unfolding is then given by:

$$\Delta G_{unfold} = \Delta G^0 + m_G[D]$$

Since we observed by previous experiments (data not shown) that the tested SHANK3$^{(1-676)}$ variants start to aggregate upon thermal ramping before the actual onset of secondary structure melting, we acquired melting curves in the presence of a non-denaturing urea concentration (2 M), which was determined from equilibrium chemical unfolding. Thermal unfolding was conducted as described previously (*Greenfield, 2007b*) with few modifications. In brief, the identical '2 M urea' sample was used, which was measured within the isothermal chemical unfolding series. CD data was acquired at 222 nm for 3 s/point with 25 μs sampling period over a temperature range of 10–90°C. Thereby, temperature changes were made in a stepped ramping mode with a heating rate of 1.0°C/min (1.0°C steps) and 30 s settling time (0.5°C tolerance). The actual sample temperature was monitored by a temperature sensor inside the sample and was used for further data processing. To determine the corresponding melting temperatures, the data was fitted to a Boltzmann model.

## Nano differential scanning fluorimetry (nanoDSF)

Differential exposure or shielding of intrinsic tryptophan reporter residues was monitored as readout for changes in protein tertiary structure upon thermal ramping by nanoDSF. True label-free fluorescence measurements were performed on a Prometheus NT.48 nanoDSF instrument (NanoTemper Technologies, Munich, Germany) using nanoDSF grade standard capillaries (10 μL, NanoTemper Technologies). Experiments were set up and monitored with the software PR.ThermControl. Before

actual measurements were performed, appropriate conditions were determined by discovery scans. Finally, measurements were conducted at 50% excitation power using a protein concentration of 0.5 mg/mL for each SHANK3$^{(1-676)}$ variant (initial $F_{330nm}$ > 5,000 fluorescence counts). Intrinsic tryptophan fluorescence emission was measured at 330 nm and 350 nm over a temperature range of 15–90°C with a heating rate of 1°C/min. Due to known attractive interparticle interactions and protein aggregation tendency from other experiments, proteins were also measured in the presence of 0.25–2.0 M urea. Nonetheless thermally induced unfolding was observed to be irreversible and accompanied with protein precipitation even in the presence of 2 M urea. $F_{330}$ and $F_{350}$ curves were blank subtracted and the ratio F(350/330 nm) was subsequently plotted against temperature. From these melting curves, the first derivatives were calculated and smoothed by a moving average with a window of 100 points. Transition points were finally determined by peak analysis of the corresponding first derivative curves.

## Intrinsic tryptophan and extrinsic ANS fluorescence spectroscopy

Fluorescence spectroscopy was performed on a F-7000 fluorescence spectrophotometer (Hitachi) using $Ni^{2+}$-IDA purified $His_6$-SUMO-SHANK3$^{(1-676)}$ variants as described above. Prior to measurement, concentrated protein stock solutions were diluted to a final concentration of 2 µM in purification buffer. Measurements were conducted at room temperature in a standard 10 mm rectangular quartz cell. For intrinsic tryptophan fluorescence emission, a wavelength scan was performed from 300 to 400 nm with a scan speed of 240 nm/min at an excitation wavelength of 295 nm (5 nm excitation and emission slit, PMT voltage set to 700 V). Protein surface hydrophobicity was measured by titrating purified $His_6$-SUMO-SHANK3$^{(1-676)}$ variants (2 µM) with 8-anilino-1-naphthalenesulfonic acid (ANS) in a concentration range of 10–100 µM. After equilibration at room temperature, ANS was excited at 365 nm and fluorescence was recorded between 400 and 700 nm with a scan speed of 240 nm/min. Finally, fluorescence spectra were Savitzky-Golay smoothed (third order, seven points window) and buffer subtracted.

## Limited proteolysis

To test the conformational stability and accessibility of individual SHANK3$^{(1-676)}$ variants to the protease trypsin, limited proteolysis was performed by incubating proteins in a molar ratio of 200:1 with trypsin-EDTA (Gibco) at room temperature. To a 1 mg/mL (13.5 µM) solution of SHANK3$^{(1-676)}$ trypsin was added to a final concentration of 67 nM in a reaction volume of 120 µL. At each timepoint (0–60 min), 15 µL of the reaction mixture were removed and added to 15 µL of 95°C preheated 2x SDS sample buffer (125 mM Tris-HCl pH 6.8, 4% [w/v] SDS, 20% [v/v] glycerol, 10% [v/v] β-mercaptoethanol, 0.004% [w/v] bromophenol blue) to facilitate an immediate stop of the proteolytic reaction. Samples were boiled at 95°C for 5 min and subsequently separated by polyacrylamide gel electrophoresis on a 4–12% pre-cast NuPage Bis-Tris protein gel (ThermoFisher Scientific). Finally, proteins were stained by Coomassie brilliant blue (CBB). Analysis of lane profiles was done in ImageJ software (NIH).

## Molecular dynamics simulations

MD simulations were performed with GROMACS 2018.4 using the Amber03 force field (*Duan et al., 2003*; *Hess et al., 2008*; *Pronk et al., 2013*). Initial coordinates were used from PDB 5G4X (*Lilja et al., 2017*). The single-point mutants R12C and L68P were created with UCSF Chimera (*Pettersen et al., 2004*). The peptides were solvated in a cubic box with periodic boundary conditions and a side length of ~110 Å (10 Å initial minimum distance of solute to all boundaries) comprising the peptide and ~43,000 $H_2O$ molecules and 2–3 chlorine ions to neutralize the protein charge. For all systems, the same molecular dynamics protocol was used. After a steepest descent energy minimization (convergence criteria 500,000 steps or maximum force <10 kJ $mol^{-1}$ $nm^{-1}$), two 100 ps equilibration MD runs were performed. The first one was performed in the constant particle number, volume, temperature ensemble (NVT; with modified Berendsen thermostat with velocity rescaling at 300 K and a 0.1 ps timestep; separate heat baths for peptide and solvent) and the second one in the constant particle number, pressure, temperature ensemble (NPT; Parrinello-Rahman pressure coupling at 1 bar with a compressibility of $4.5 \times 10^{-5}$ $bar^{-1}$ and a 2 ps time constant; *Bussi et al.,*

*2007*; *Nosé and Klein, 1983*; *Parrinello and Rahman, 1981*). During each equilibration run, a position restraint potential with a force constant of 1000 kJ mol$^{-1}$ nm$^{-2}$ was added to all peptide atoms.

For MD simulations the leap-frog integrator was used with a time step of 2 fs. Coordinates were saved every 10 ps. The same temperature and pressure coupling schemes as applied for the equilibration runs were used for the subsequent MD simulations. All bonds to hydrogen atoms were constrained using the Linear Constrained Solver (LINCS) with an order of 4 and one iteration (*Hess et al., 1997*). A grid-based neighbor list with a threshold of 10 Å was used and updated every five steps (10 fs). The particle-mesh Ewald method was used for long-range electrostatic interactions above 10 Å with a fourth-order interpolation and a maximum spacing for the FFT grid of 1.6 Å (*Darden et al., 1993*; *Essmann et al., 1995*). Lennard-Jones interactions were cut-off above 10 Å. A long-range dispersion correction for energy and pressure was used to compensate for the Lennard-Jones interaction cut-off (*Hess et al., 2008*).

MD trajectories were visualized and analyzed using VMD 1.9.3 (*Humphrey et al., 1996*). Backbone RMSD and RMSF traces were calculated from every fifth frame using the inbuilt RMSD Trajectory Tool or a Tcl script, respectively.

## Preparation of primary hippocampal neurons and transfection

Cultures of primary hippocampal neurons were prepared as described previously (*van Bommel et al., 2019*). Pregnant Wistar rats Crl:WI (Charles River; E18) were sacrificed and hippocampi were dissected from E18 embryos. Following 10 min of trypsin treatment at 37°C, hippocampi were physically dissociated and plated with a density of 30,000 cells/mL on 18 mm glass coverslips, which have been coated with poly-L-lysine (PLL). Cells were initially plated in DMEM supplemented with 10% (v/v) fetal bovine serum (FBS) as well as Penicillin/Streptomycin (PS). After one hour, the medium was exchanged with BrainPhys neuronal medium supplemented with SM1 and 0.5 mM glutamine. Neurons were grown and maintained at 37°C, 5% CO$_2$, and 95% humidity. For transfection, both plasmid DNA and lipofectamine 2000 were diluted accordingly in blank BrainPhys neuronal medium, mixed (DNA/lipofectamine ratio 1:3), and incubated at room temperature for 40 min. In parallel, the neuronal growth medium was collected and replaced by BrainPhys medium supplemented with 0.5 mM glutamine. Subsequently, the transfection mix was added to the neurons for 1–1.5 hr before the medium was exchanged back to the conditioned medium. Expression periods were limited to <24 hr.

## Immunostainings, spinning disk confocal microscopy, and image analysis

Cultured rat primary hippocampal neurons were fixed in 4% paraformaldehyde, 4% sucrose, in PBS for 10 min at room temperature. Subsequently, cells were washed three times with PBS and permeabilized in 0.2% Triton X-100 in PBS for 10 min. After three washes with PBS, neurons were incubated in blocking buffer (BB, 10% horse serum, 0.1% Triton X-100 in PBS) for 45 min at room temperature and subjected to antibody staining. Incubation with primary antibodies was done in BB at 4°C overnight. Following three washes in PBS, neurons were incubated with corresponding secondary antibodies in BB for 2 hr at room temperature. Finally, cells were washed three times in PBS and mounted on microscopy slides with Mowiol (Carl Roth; prepared according to manufacturer's protocol including DABCO as antifading agent).

Spinning-disk confocal microscopy was performed with a Nikon Eclipse Ti-E microscope. The microscope was controlled by the VisiView software (VisitronSystems) and equipped with 488, 561 and 639 nm excitation lasers coupled to a CSU-X1 spinning-disk unit (Yokogawa) via a single-mode fiber. Emission was collected through a quad-band filter (Chroma, ZET 405/488/561/647 m) on an Orca flash 4.0LT CMOS camera (Hamamatsu). For fixed primary hippocampal neurons, Z-stack images were taken by using a 100x objective (Nikon, ApoTIRF 100×/1.49 oil). The pixel size was set as 65 nm$^2$ and Z-stack interval was 0.3 μm.

All images were processed and analyzed using ImageJ. For co-localization analysis of either endogenous SHANK3 or overexpressed GFP-SHANK3$^{FL}$ variants (WT, R12C or L68P) with synaptic markers (homer and bassoon), the shape of the entire analyzed dendrite, including spines, was drawn based on MAP2 and synaptic marker staining by hand using the segmented line tool. Subsequently, the number of SHANK3, homer, bassoon as well as co-localized puncta was detected via

the ComDet v.0.5.1 plugin. The fraction of co-localization was calculated by dividing the amount of co-localized puncta by the total number of SHANK3, homer and bassoon puncta, respectively.

For the quantification of spine density and SHANK3 clusters in neurons overexpressing GFP-SHANK3$^{FL}$ variants, mRuby2 was co-transfected in all three conditions as a volume marker. A construct expressing GFP was used as control to monitor the dosage-dependent effect of SHANK3 overexpression. The number of spines was counted using the multi-point tool based on the mRuby2 channel and normalized to a dendritic length of 10 μm.

The number of SHANK3 clusters in the dendrite (SHANK3$_{den}$) as well as in spines (SHANK3$_{spi}$) was quantified for each genotype using the multi-point tool. The fraction of SHANK3 clusters in dendrite versus spines was then calculated for each genotype by individually dividing SHANK3$_{den}$ and SHANK3$_{spi}$ by the total number of SHANK3 clusters.

For the fluorescence intensity analysis of SHANK3 clusters in dendrite and spines, a line profile covering the entire spine width was drawn based on the mRuby2 and SHANK3 channel. The line profile initiates at the head of a measured spine and terminates on its adjacent dendrite. Subsequently, the mean gray value was plotted along the line profile. Finally, detected peak values from the spine and dendrite region were divided by each other to calculate the ratio of SHANK3 cluster distribution.

## Co-immunoprecipitation (co-IP) and western blotting

Proteins expressed in HEK 293 T cells were used for co-IP. HEK 293 T cells are regularly tested for mycoplasma contamination using PCR-based testing. The cell line is additionally verified periodically by morphological criteria and checked for the presence of the neomycin resistance gene. Cells are cultured for a maximum of 20 passages and then discarded.

HEK 293 T cells were co-transfected with C-terminally RFP-tagged Shank3$^{FL}$ constructs (WT, R12C, L68P) and constructs coding for GFP-tagged interaction partners (SAPAP1, homer1, or cortactin) using the TurboFect transfection reagent (ThermoFisher Scientific). The next day, cells were lysed in radioimmuno precipitation assay (RIPA) buffer (50 mM Tris-HCl pH 8.0, 150 mM NaCl, 1% (v/v) NP40, 0.5% (w/v) sodium deoxycholate, 5 mM EDTA, 0.1% (w/v) SDS), and centrifuged at 20,000 g for 20 min at 4°C. Subsequently, RFP-tagged proteins were immunoprecipitated from the supernatant using 20 μL of RFP-trap agarose matrix (Chromotek, Munich, Germany) for 2 hr at 4°C on a rotator. The agarose matrix was then collected by centrifugation at 1000 g for 2 min and washed five times with RIPA buffer, followed each time by centrifugation.

For western blotting, input (cell lysate) and precipitate samples were denatured in 1x Laemmli sample buffer (63 mM Tris-HCl pH 6.8, 10% [v/v] glycerol, 1.5% [w/v] SDS, 100 mM dithiothreitol and 0.01% [w/v] bromophenol blue) at 95°C, separated by SDS-PAGE at 100–180 V and transferred to a nitrocellulose membrane in transfer buffer (25 mM Tris, 192 mM glycine, 0.05% [w/v] SDS and 20% [v/v] methanol) at 100V for 100 min using a MINI PROTEAN II system (Bio-Rad). Membranes were subsequently blocked with 5% (w/v) milk powder in TBS-T (Tris buffered saline; 10 mM Tris-HCl pH 8.0, 150 mM NaCl, supplemented with 0.05% [v/v] Tween 20) and incubated with primary antibodies overnight at 4°C, followed by HRP-linked secondary antibodies at room temperature for 1 hr. After washing the membranes with TBS-T, chemiluminescence was detected using the WesternBright chemiluminescence substrate (Biozym, Hess. Oldendorf, Germany). Finally, membranes were scanned using a ChemiDoc MP imaging system (Bio-Rad) and images were processed and analyzed using the Image Lab software (Bio-Rad).

## Fluorescence recovery after photobleaching (FRAP)

To measure SHANK3 dynamics in individual spines, cultures of primary rat hippocampal neurons were co-transfected at DIV14–16 with 1.8 μg of pHAGE2-EF1a-GFP-SHANK3$^{FL}$ (WT or carrying one of the studied mutations) and 0.5 μg of pAAV-mRuby2 for 16–18 hr to avoid morphological artifacts of SHANK3 overexpression. Live imaging and FRAP experiments were performed on the imaging system described above, using a 100x objective (CFI Apochromat TIRF 100XC oil, 1.49 NA). Imaging settings including FRAP were defined within the software VisiView (Visitron Systems, Puchheim, Germany). After five baseline images (16-bit, 65 nm$^2$/px) taken with 5 s interval, photobleaching was achieved by scanning each ROI (individually selected spines) with 2 ms/pixel at 488 nm (50–70% laser power). Subsequently, 120 post-bleach images were acquired with 5 s interval. For stimulation of

neuronal activity, cultures were incubated for 5 min at 5% $CO_2$ and 37°C with a final concentration of 50 µM bicuculline prior to imaging. In the presence of bicuculline, live imaging and photobleaching was continued for a maximum of 45 min to avoid artifacts due to neuronal over-activation.

For measurement of SHANK3 interaction partner dynamics (cortactin, homer1), cultures of primary rat hippocampal neurons were co-transfected at DIV14-16 with 1.0 µg of pHAGE2-EF1a-GFP-SHANK3[FL] (WT or mutant) and 1.0 µg of either pFUGW-f(syn)homer1-tdTomato-WPRE or mCherry-cortactin for 16–18 hr. Live imaging and FRAP experiments were performed as described above, except for photobleaching, which was achieved by scanning each ROI with 3 ms/pixel at 561 nm (100% laser power).

Fluorescence values were obtained from each image using ImageJ. Therefore, ROIs were drawn on bleached spines, non-bleached control regions, and the background. Mean gray values were obtained for each frame by using the 'plot z-axis profile' function within ImageJ. Subsequently, FRAP traces for each spine were background subtracted, normalized to the non-bleached control, and scaled between 0 and 1. Statistical analysis of all imaging data was performed with the Origin 2019b software package.

### Fluorescent non-canonical amino acid tagging (FUNCAT)

For direct in situ visualization of newly synthesized proteins in hippocampal primary neurons overexpressing GFP-SHANK3[FL] variants, FUNCAT was performed as previously described with few adaptations (*Dieterich et al., 2010*). Briefly, to account for the neuronal activity dependence of de novo protein synthesis, one group of transfected neurons (DIV 15–17, <24 hr of overexpression) was pharmacologically silenced by treatment with 1 µM tetrodotoxin (TTX) and 50 µM D-(-)-2-amino-5-phosphonopentanoic acid (D-APV) prior to metabolic labeling. Transfected neurons were then metabolically labeled by replacing the growth medium with methionine-free HibA medium (BrainBits LLC) for 20 min to deplete endogenous methionine. Subsequently, cells were incubated for 2–3 hr at 37°C, 5% $CO_2$ in HibA medium supplemented either with 4 mM L-azidohomoalanine (AHA; Click Chemistry Tools) or 4 mM L-methionine as control. Subsequently, neurons were washed with cold PBS-MC (1 mM $MgCl_2$, 0.1 mM $CaCl_2$ in PBS, pH 7.4) on ice and fixed with 4% paraformaldehyde, 4% sucrose for 10 min in PBS at room temperature (RT; 20–25°C). Following three washing steps with PBS (pH 7.4), fixed cells were permeabilized with PBS containing 0.25% Triton X-100 for 10 min, washed again three times with PBS, and incubated with blocking solution (10% horse serum, 0.1% Triton X-100 in PBS) for 1 hr at RT. Blocked cells were finally washed three times with PSB (pH 7.8).

For click-labeling of the AHA-modified protein pool by copper-catalyzed azide-alkyne [3 + 2] cycloaddition (CuAAC), a reaction mix containing 200 µM Tris-[(1-benzyl-1H-1,2,3-triazol-4-yl)-methyl]-amine (TBTA), 500 µM Tris(2-carboxyethyl)phosphine hydrochloride (TCEP), 2 µM fluorescent alkyne tag (TAMRA-PEG4-alkyne), and 200 µM $CuSO_4$ was prepared in PBS (pH 7.8). After each addition of a reagent, the reaction mix was vortexed for several seconds. Stock solutions for each reagent except TBTA were prepared in sterile water, TBTA was dissolved in DMSO. Hippocampal primary neurons were incubated with the reaction mix in a dark, humidified box overnight at RT with gentle agitation. Cells were subsequently washed three times with FUNCAT wash buffer (0.5 mM EDTA, 1% Tween-20 in PBS, pH 7.8) followed by two additional washes with PBS (pH 7.4).

Following click-labeling, neurons were immunostained for homer as described above to allow detection of labeled and unlabeled excitatory spines. Finally, labeled and immunostained cells were mounted on microscopy slides with Mowiol and imaged with spinning-disk confocal microscopy as described above. Analysis of synaptic TAMRA intensities was done in ImageJ and data were plotted and analyzed in Origin 2019b.

## Acknowledgements

We would like to thank Friederike Schröder for help with FUNCAT experiments, Dr. Rajeev Raman for help with fluorescence spectroscopy measurements, Dr. Maria Garcia Alai and Dr. Rob Meijers for the access to SPC facility EMBL Hamburg and for very helpful advice regarding experimental design and instruments operation. We thank Dr. Kim Remans (EMBL Heidelberg) for providing the pETM11-SenP2 plasmid and Dr. Thorsten Trimbuch from the Viral core facility (vcf.charite.de) for providing f(syn)Homer1-tdTomato-wpre plasmid (BL-1034/in a collaboration with DFG SFB1315 C01). We also thank Dr. Julia Bär, Dr. Maria Andres-Alonso, Dr. Katarzyna Grochowska, and Daniela

Hacker for contributing to the preparation of primary rat hippocampal neuron cultures. Furthermore, we thank Dr. Jasper Grendel for help with analysis. This work was supported by the DAAD Research Stays for University Academics and Scientists Award to ASK; Leibniz Pakt für Forschung und Innovation 'Neurotranslation' to EJK, MRK and MM; the iNEXT MX/SAXS ES (PID: 5843), the Deutsche Forschungsgemeinschaft (DFG Emmy Noether Programme MI1923/1-2, FOR2419 TP2, and Excellence Strategy—EXC-2049–390688087), Hamburg Landesforschungsförderung LFF-FV76, Hertie Network of Excellence in Clinical Neuroscience and Excellence Strategy Program to MM, DFG FOR2419 TP3, DFG CRC 1436 Project A02, DFG RTG 2413 SynAge TP3 and 4 to MRK, DFG Kr1321/9-1 to HJK and Bundesministerium für Bildung und Forschung (BMBF) 16QK10A (SAS-BSOFT) to DM.

## Additional information

### Funding

| Funder | Grant reference number | Author |
|---|---|---|
| Deutsche Forschungsgemeinschaft | MI1923/1-2 | Marina Mikhaylova |
| Deutsche Forschungsgemeinschaft | FOR2419 TP2 | Marina Mikhaylova |
| Deutsche Forschungsgemeinschaft | EXC-2049-390688087 | Marina Mikhaylova |
| Leibniz-Gemeinschaft | Neurotranslation | Eunjoon Kim Michael R Kreutz Marina Mikhaylova |
| Deutscher Akademischer Austauschdienst | Research Stays for University Academics and Scientists Award | Alla S Kostyukova |
| University of Hamburg | LFF-FV76 | Marina Mikhaylova |
| Deutsche Forschungsgemeinschaft | FOR2419 TP3 | Michael R Kreutz |
| Deutsche Forschungsgemeinschaft | CRC 1436 Project A02 | Michael R Kreutz |
| Deutsche Forschungsgemeinschaft | RTG 2413 SynAge TP3 and TP4 | Michael R Kreutz |
| Deutsche Forschungsgemeinschaft | Kr1321/9-1 | Hans-Jürgen Kreienkamp |
| Bundesministerium für Bildung und Forschung | 16QK10A (SAS-BSOFT) | Dmitry Molodenskiy |
| Deutsche Forschungsgemeinschaft | Kr1321/9-1 | Hans-Jürgen Kreienkamp |

The funders had no role in study design, data collection and interpretation, or the decision to submit the work for publication.

### Author contributions

Michael Bucher, Conceptualization, Data curation, Formal analysis, Investigation, Visualization, Methodology, Writing - original draft, Writing - review and editing; Stephan Niebling, Resources, Software, Investigation, Methodology; Yuhao Han, Formal analysis, Investigation, Visualization; Dmitry Molodenskiy, Data curation, Investigation, Methodology; Fatemeh Hassani Nia, Investigation; Hans-Jürgen Kreienkamp, Resources; Dmitri Svergun, Methodology, Provided instrumentation for SAXS; Eunjoon Kim, Funding acquisition, Writing - review and editing; Alla S Kostyukova, Data curation, Supervision, Methodology; Michael R Kreutz, Conceptualization, Resources, Funding acquisition, Methodology, Writing - review and editing; Marina Mikhaylova, Conceptualization, Resources, Data curation, Supervision, Funding acquisition, Project administration, Writing - review and editing

## Author ORCIDs

Michael Bucher https://orcid.org/0000-0001-9296-121X
Dmitry Molodenskiy http://orcid.org/0000-0002-5954-4294
Hans-Jürgen Kreienkamp http://orcid.org/0000-0002-8871-9970
Marina Mikhaylova https://orcid.org/0000-0001-7646-1346

## Ethics

Animal experimentation: All animal experiments were carried out in accordance with the European Communities Council Directive (2010/63/EU) and the Animal Welfare Law of the Federal Republic of Germany (Tierschutzgesetz der Bundesrepublik Deutschland, TierSchG) approved by the local authorities of the city-state Hamburg (Behörde für Gesundheit und Verbraucherschutz, Fachbereich Veterinärwesen) and the animal care committee of the University Medical Center Hamburg-Eppendorf.

## Decision letter and Author response

Decision letter https://doi.org/10.7554/eLife.66165.sa1
Author response https://doi.org/10.7554/eLife.66165.sa2

# Additional files

## Supplementary files

• Supplementary file 1. Principal SAXS parameters computed for indicated $His_6$-SUMO-SHANK3$^{(1-676)}$ variants.

• Transparent reporting form

## Data availability

All data generated or analysed during this study are included in the manuscript and supporting files.

The following datasets were generated:

| Author(s) | Year | Dataset title | Dataset URL | Database and Identifier |
|---|---|---|---|---|
| Bucher M, Molodenskiy D, Svergun D | 2021 | SASDLJ3 - SH3 and multiple ankyrin repeat domains protein 3 (wild type) | https://www.sasbdb.org/data/SASDLJ3/ | SASBDB, SASDLJ3 |
| Bucher M, Molodenskiy D, Svergun D | 2021 | SASDLK3 - SH3 and multiple ankyrin repeat domains protein 3 with a point mutation (L68P) | https://www.sasbdb.org/data/SASDLK3/ | SASBDB, SASDLK3 |
| Bucher M, Molodenskiy D, Svergun D | 2021 | SASDLL3 - SH3 and multiple ankyrin repeat domains protein 3 with a point mutation (R12C) | https://www.sasbdb.org/data/SASDLL3/ | SASBDB, SASDLL3 |

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
