## [Decision Letter]

**Acceptance summary:**

The post-synaptic protein SHANK3 has a critical role in the function of synapses and its dysfunctions, for example arising from missense mutations, is associated with neurological disorders. As often the case, the molecular principles underlying the exact role of such mutations on the structure and function of proteins, remains to be resolved experimentally. Missense mutations could impact on protein folding or stability and thus its availability, on protein structure or its dynamics, and on protein interactions, but their effects could be very subtle. The authors use an extensive set of biochemical, biophysical and cellular tools to show that two autism associated missense mutations in SHANK3 affect the structure, dynamics and its availability at the synapse in different ways and on different structural levels. These alterations modify protein folding, alter synaptic targeting and changes protein turnover at synaptic sites. This paper increases our knowledge about the relation between the molecular functional alteration in mutated SHANK3 and the pathogenesis of autism spectrum disorder associated with these mutations.

**Decision letter after peer review:**

Thank you for submitting your article "Autism associated SHANK3 missense point mutations impact conformational fluctuations and protein turnover at synapses" for consideration by *eLife*. Your article has been reviewed by 2 peer reviewers, and the evaluation has been overseen by a Reviewing Editor and Olga Boudker as the Senior Editor. The reviewers have opted to remain anonymous.

Essential revisions:

1. The authors declare that only about 70% of glutamatergic spines will be affected by the two SHANK3 missense mutations. However, this is based on the staining of endogenous Shank3 and thus is depending on the efficiency of the antibodies used that might either recognize only some Shank3 splice variants or are not sufficiently strong to recognize spines containing an amount of Shank3 lower than the antibodies threshold sensitivity. Secondly, it's unclear whether these data have a real relevance related to the general neuronal function altered in ASD, 70% of glutamatergic spines are the majority of the spines. Thus it will be important to provide stronger evidence that Shank3 is absent in some spines or to eliminate the sentence that only 70% of glutamatergic spines will be affected by the two SHANK3 missense mutations.

2. Data presented in Figure 5 suggest that the mutations lead to an increased fraction of dendritic shaft SHANK3 clusters. However, it's not clear if these are or are not synaptic clusters. How much of these clusters colocalize with presynaptic markers? Do these dendritic SHANK3 clusters colocalize with Homer?

3. The following sentence: "…this study, we bridge structural data with the corresponding synaptic phenotype" is not fully correct. Indeed there is no data that molecularly links the structural data with all the synaptic altered phenotype. For example non of the structural data can explain the altered turnover of mutated Shank3. Thus it will be more correct to state that the described structural data potentially correlate with the altered synaptic phenotype but they can be totally independent. Alternatively, the authors should provide more data that demonstrate the direct links between the two types of observed sets of data. But this might not be the scope of the work.

4. The data that shows the increased synaptic turnover of the R12C mutant is interesting, but t will be more interesting if the authors demonstrate that Shank3 binding partners turnover is also alerted. It is possible for example to study the turnover of synaptic WT Shank or Homer in the presence of the R12C mutant?

5. Increased synaptic turnover of the R12C mutant might or might not depend on the altered interaction of the mutant with Shank3 principal protein partners like GKAP or Homer. Thus, with a set of immunoprecipitation experiments, it might be possible to study if the mutants interact with less affinity with GKAP or Homer.

6. Possibly the SPN-ARR interface is destabilized by the L68P mutation and the interface between these two domains is disrupted due to the L68P mutation, resulting in the increased flexibility as seen in the SAXS data. This should be tested in the CORAL modelling by treating the SPN and ARR domains as two separate rigid bodies, connected by flexible linker.

7. The flexibility in the proteins, as observed in the SAXS data, should caution the authors in interpreting their static CORAL models as is now discussed (e.g. lines 126-130 and lines 317-324). Perhaps the authors can focus their interpretation more on the additional flexibility for the L68P mutant observed in the SAXS data, which is striking and solid data, and less on the static CORAL models?

8. Figure 1C-F and Suppl. Figure 3B. Dmax is concentration dependent for all variants. How certain are the authors that the extrapolated data represents monomeric protein? For example, comparing the lowest concentration data for all variants, the Dmax is largest for the wt protein, while for the extrapolated data the wt protein has the shortest Dmax. In addition, the Dmax for the extrapolated data is larger than the Dmax for the lowest concentration data. If the extrapolated data represents a fraction of dimers, how does this influence the CORAL modelling?

9. The outcome of the molecular dynamics simulation may be different if the start structure is adjusted, i.e. if the simulation is done with a start model that has the two domains separated. This should be tested to confirm that the SPN-ARR interaction is not affected by the mutations.

10. The experimental study of disease related missense mutations on protein structure and stability is not new. See for examples doi.org/10.1074/jbc.M210019200 and 10.1371/journal.pone.0186110. Also, in SHANK3 another disease related missense mutation (R536W) has been investigated structurally doi.org/10.1111/jnc.14322. I would like to suggest that lines 299 and 300 are changed or removed to reflect this better. In addition, the "In a proof-of-principle study" in the abstract may be misleading taking this previous work into account.

11. It would strengthen the discussion if the authors could discuss the cellular readout reported here and by others more in the light of their results on the flexibility and stability of the mutants.

*Reviewer #1:*

In this paper, Bucher et al. demonstrate that two Shank3 ASD-associated point mutations show distinct changes in the secondary and tertiary structure of the scaffold protein as well as higher conformational fluctuations. These alterations modify protein folding, alter synaptic targeting and changes protein turnover at synaptic sites.

This is a very interesting paper that increases our knowledge about the relation between the molecular functional alteration in mutated SHANK3 and the pathogenesis of ASD associated with these mutations.

Most of the experiments are well presented and convincible performed. However, the following points should be addressed in order to improve the strength of the proposed findings.

1. The authors declare that only about 70% of glutamatergic spines will be affected by the two SHANK3 missense mutations. However, this is based on the staining of endogenous Shank3 and thus is depending on the efficiency of the antibodies used that might either recognize only some Shank3 splice variants or are not sufficiently strong to recognize spines containing an amount of Shank3 lower than the antibodies threshold sensitivity. Secondly, it's unclear whether these data have a real relevance related to the general neuronal function altered in ASD, 70% of glutamatergic spines are the majority of the spines. Thus it will be important to provide stronger evidence that Shank3 is absent in some spines or to eliminate the sentence that only 70% of glutamatergic spines will be affected by the two SHANK3 missense mutations.

2. Data presented in figure 5 suggest that the mutations lead to an increased fraction of dendritic shaft SHANK3 clusters. However, it's not clear if these are or are not synaptic clusters. How much of these clusters colocalize with presynaptic markers? Do these dendritic SHANK3 clusters colocalize with Homer?

3. The following sentence: "…this study, we bridge structural data with the corresponding synaptic phenotype" is not fully correct. Indeed there is no data that molecularly links the structural data with all the synaptic altered phenotype. For example non of the structural data can explain the altered turnover of mutated Shank3. Thus it will be more correct to state that the described structural data potentially correlate with the altered synaptic phenotype but they can be totally independent. Alternatively, the authors should provide more data that demonstrate the direct links between the two types of observed sets of data. But this might not be the scope of the work.

4. The data that shows the increased synaptic turnover of the R12C mutant is interesting, but t will be more interesting if the authors demonstrate that Shank3 binding partners turnover is also alerted. It is possible for example to study the turnover of synaptic WT Shank or Homer in the presence of the R12C mutant?

5. Increased synaptic turnover of the R12C mutant might or might not depend on the altered interaction of the mutant with Shank3 principal protein partners like GKAP or Homer. Thus, with a set of immunoprecipitation experiments, it might be possible to study if the mutants interact with less affinity with GKAP or Homer.

*Reviewer #2:*

The post-synaptic protein SHANK3 has a critical role in the function of synapses and its dysfunctions, for example arising from missense mutations, is associated with neurological disorders. As often the case, the molecular principles underlying the exact role of such mutations on the structure and function of proteins, remains to be resolved experimentally. Missense mutations could impact on protein folding or stability and thus its availability, on protein structure or its dynamics, and on protein interactions, but their effects could be very subtle. The authors use an extensive set of biochemical, biophysical and cellular tools to show that two missense mutations in SHANK3 affect the structure, dynamics and its availability at the synapse in different ways and on different structural levels.

The effect of the two mutants, R12C and L68P is investigated at different structural levels. SHANK3 is a multidomain protein in which several domains are connected by flexible linkers. This makes detailed structural studies on this protein challenging. Small angle X-ray scattering (SAXS) is very suited to obtain low-resolution structural data on such flexible proteins. The authors show that the L68P SHANK3 mutant, in the context of a construct consisting of residues 1-676 fused N-terminally to a SUMO domain, has more intrinsic flexibility, whereas the R12C mutant does not seem to be affected in its flexibility compared to wt SHANK3. By modelling, using previously reported high-resolution structural data for the individual domains SUMO, SH3 and PDZ and the combination SPN-ARR as one unit, and assuming flexible linkers between these rigid bodies, the additional flexibility in the L68P variant is attributed to the decoupling of the SH3 and PDZ domains from the ARR domain. This is surprising, as the mutation, L68P, is located in the SPN domain, near to the ARR domain but far away from the SH3 and PDZ domains in all the CORAL models. Possibly the SPN-ARR interface is destabilized by the L68P mutation and the interface between these two domains is disrupted due to the L68P mutation, resulting in the increased flexibility. While such an interpretation could perhaps also explain the data, this is not considered. In addition, the flexibility in the proteins, as observed in the SAXS data, should caution the interpretation of the static CORAL models. The increased flexibility of the L68P mutant is however striking and clear-cut from the data.

The molecular dynamics simulations using the SPN-ARR combination as a template do not reveal destabilization of this interface by the L68P mutation. However, this may be because the start model in the simulation already assumes that both domains are interacting. The outcome of the molecular dynamics simulation could be different if the start structure is adjusted, i.e. if the simulation is done with a start model that has the two domains separated. Such analysis may also help in supporting the modelling of the SAXS data.

The influence of the mutations on the thermal, chemical and proteolytic stability of the proteins is thoroughly probed by tryptophan and ANS fluorescence spectroscopy to reveal tertiary level changes, by circular dichroism to visualize secondary level changes and by limited proteolysis to reveal accessibility changes of proteolytic sites. The tryptophan fluorescence spectroscopy (nDSF) shows that the L68P mutant has reduced tertiary structure stability. It is however not straightforward to interpret the data due to multiple transition events and precipitation of unfolded proteins. The authors use urea and classification of the data in three temperature transition zones to tease out some information. But is not clear how reproducible this data is. The overall conclusion that the L68P mutant has reduced tertiary structural stability, and that the R12C mutant is not affected on this level is however supported by the data and is also verified by subsequent extrinsic fluorescence emission spectroscopy. Monitoring thermal unfolding at the secondary structure paints a different picture compared to the tertiary structures and indicates that the R12C mutant is more stable compared to the L68P and wt SHANK3. Also limited proteolysis indicates the R12C mutant is a bit more stable.

While the mutations do not seem to affect protein production, i.e. equal quantities are produced compared to wt when overexpressed in hippocampal neurons, the availability of the SHANK3 variants may differ. One of the two mutants, R12C, has an increased recovery in spines associated with a reduced residing time and, thus possibly, less synaptic SHANK3. The availability of the other mutant L68P, may be decreased as it seems to cluster more in dendrites, compared to wt protein. These are interesting findings and this altered availability may explain their association with autism spectrum disorder.

Both the cellular readouts, i.e. increased diffusion of R12C and increased clustering of L68P, may be the result of altered protein interactions. However, this is not looked at further experimentally in this work but has been previously reported and is clearly described in the introduction and discussion of this manuscript. In addition, it is not clear if the cellular findings reported here are related to the differential stabilities of the proteins, i.e. L68P SHANK3 less stable and R12C SHANK3 more stable, nor to the increased flexibility of L68P SHANK3. A more thorough discussion of the cellular readout reported here and by others in the light of the results on the flexibility and stability of the mutants would strengthen the discussion of the paper.

---

## [Author Response]

Essential revisions:1. The authors declare that only about 70% of glutamatergic spines will be affected by the two SHANK3 missense mutations. However, this is based on the staining of endogenous Shank3 and thus is depending on the efficiency of the antibodies used that might either recognize only some Shank3 splice variants or are not sufficiently strong to recognize spines containing an amount of Shank3 lower than the antibodies threshold sensitivity. Secondly, it's unclear whether these data have a real relevance related to the general neuronal function altered in ASD, 70% of glutamatergic spines are the majority of the spines. Thus it will be important to provide stronger evidence that Shank3 is absent in some spines or to eliminate the sentence that only 70% of glutamatergic spines will be affected by the two SHANK3 missense mutations.

We would like to thank the reviewers for this valuable comment. Indeed, considering the large number of Shank3 isoforms it is possible that the antibody does not recognize all of them. Therefore, the number might be higher than 70%. We rephrased the sentence stating that only 70% of glutamatergic spines will be affected by the two SHANK3 missense mutations and instead suggest now that it is the majority of spines.

2. Data presented in Figure 5 suggest that the mutations lead to an increased fraction of dendritic shaft SHANK3 clusters. However, it's not clear if these are or are not synaptic clusters. How much of these clusters colocalize with presynaptic markers? Do these dendritic SHANK3 clusters colocalize with Homer?

We think this is a very important point. To address this issue, we therefore performed additional experiments in rat hippocampal neurons, transfected with GFP-SHANK3^FL^ (WT, R12C and L68P), fixed and stained for endogenous homer or bassoon as post- and presynaptic markers. We used MAP2 as a mask for the dendritic shaft and quantified the colocalization of dendritic SHANK3-clusters with homer and bassoon. Interestingly, we could see again that the L68P variant forms more clusters in the dendrite but without a difference in the fraction of synaptic clusters between genotypes. Thus, approximately 50% of clusters of both mutants and the WT were in apposition with a presynaptic terminal but only one third contained homer1, which is remarkably different as compared to spine synapses (almost 100%). However, since the L68P variant forms a higher absolute number of dendritic SHANK3clusters, an increased number of shaft synapses are present in each neuron. These data are included in Figure 5F-I (main manuscript). In the discussion we address that this analysis indirectly suggests that the composition of SHANK3-positive synapses might be different in shaft as compared to spine synapses.

3. The following sentence: "…this study, we bridge structural data with the corresponding synaptic phenotype" is not fully correct. Indeed there is no data that molecularly links the structural data with all the synaptic altered phenotype. For example non of the structural data can explain the altered turnover of mutated Shank3. Thus it will be more correct to state that the described structural data potentially correlate with the altered synaptic phenotype but they can be totally independent. Alternatively, the authors should provide more data that demonstrate the direct links between the two types of observed sets of data. But this might not be the scope of the work.

We agree with the reviewers and removed this statement. Instead we now state that “in this work we found that missense mutation-induced impairments of the structural integrity of SHANK3 serve as molecular starting point for higher order pathogenic processes associated with ASD”.

4. The data that shows the increased synaptic turnover of the R12C mutant is interesting, but t will be more interesting if the authors demonstrate that Shank3 binding partners turnover is also alerted. It is possible for example to study the turnover of synaptic WT Shank or Homer in the presence of the R12C mutant?

We performed this experiment and analyzed the turnover of two prominent interaction partners, homer1 and cortactin, in the presence of ASD-associated SHANK3 mutants. We found that the overall recovery and kinetics of homer1 was not changed upon overexpression of both mutants. However, mutations impacted the kinetics and overall synaptic turnover of cortactin, another binding partner of SHANK3 and a prominent synaptic actin-binding protein. This is a very interesting finding because it links ASD-associated SHANK3 mutations potentially to spinous actin dynamics (Macgillavry et al., 2016; Mikhaylova et al., 2018). The new data are included in Figure 7B, C, E and F (main manuscript).

5. Increased synaptic turnover of the R12C mutant might or might not depend on the altered interaction of the mutant with Shank3 principal protein partners like GKAP or Homer. Thus, with a set of immunoprecipitation experiments, it might be possible to study if the mutants interact with less affinity with GKAP or Homer.

We want to thank the reviewers for this suggestion. We performed coimmunoprecipitation experiments for SHANK3 mutants and major interaction partners, namely GKAP (SAPAP1), homer1 and cortactin and found no apparent differences in binding efficiency. These data are included Figure 6 (main manuscript). These findings also served as starting point for performing FRAP experiments requested in point #4 of Essential Revisions.

6. Possibly the SPN-ARR interface is destabilized by the L68P mutation and the interface between these two domains is disrupted due to the L68P mutation, resulting in the increased flexibility as seen in the SAXS data. This should be tested in the CORAL modelling by treating the SPN and ARR domains as two separate rigid bodies, connected by flexible linker.

To address the reviewer’s question we performed CORAL fits of zero-extrapolated SAXS data, where the linker region between the SPN- and ARR-domain has been replaced *in silico* with flexible dummy residues to capture potential disruptions of the interface. In this analysis we observed that the SPN-domain is decoupled from the ARR-domain in both mutants, suggesting a disruption of the domain interface (Figure 1F, main manuscript). However, we additionally performed a dedicated ensemble optimization method (EOM) to characterize protein flexibility numerically (see Author response image 1). Here we found no statistically significant differences between mutants and the WT. We therefore decided to not to include the data in the manuscript.

**Author response image 1. respfig1:** SAXS EOM analysis of His6-SUMO-SHANK3(1-676) protein flexibility. (from left to right)The ensemble distributions of the maximum particle diameter (D_max_) as well as the radius of gyration (R_g_) are shown for His_6_SUMO-SHANK3^(1-676)^ variants. Differences in the degree of flexibility (R_flex_) of selected sub-ensembles between the WT and mutants are in the range of 1.5 – 4%, which is negligible.

7. The flexibility in the proteins, as observed in the SAXS data, should caution the authors in interpreting their static CORAL models as is now discussed (e.g. lines 126-130 and lines 317-324). Perhaps the authors can focus their interpretation more on the additional flexibility for the L68P mutant observed in the SAXS data, which is striking and solid data, and less on the static CORAL models?

While addressing point #6 of essential revisions, we found no significant differences in protein flexibility by EOM (please see Author response image 1). We are aware that a Kratky plot provides qualitative information on protein folding and flexibility. Based on the shape of the Kratky plot for the L68P mutant we conclude partial unfolding of the mutant which is further supported by CORAL analysis as well as intrinsic and extrinsic fluorescence spectroscopy data. We think, however, in this case folding and flexibility should be interpreted separately.

8. Figure 1C-F and Suppl. Figure 3B. Dmax is concentration dependent for all variants. How certain are the authors that the extrapolated data represents monomeric protein? For example, comparing the lowest concentration data for all variants, the Dmax is largest for the wt protein, while for the extrapolated data the wt protein has the shortest Dmax. In addition, the Dmax for the extrapolated data is larger than the Dmax for the lowest concentration data. If the extrapolated data represents a fraction of dimers, how does this influence the CORAL modelling?

We want to thank the reviewers for pointing this out. To address this question, we performed OLIGOMER-analysis of SAXS data spanning the entire measured concentration range including zero-extrapolated SAXS profiles to estimate the concentration-dependency of the monomeric SHANK3 volume fraction. We found that zero-extrapolated data indeed represents almost exclusively monomeric protein (Figure 1—figure supplement 3C). Therefore, dimers will not influence the CORAL modelling. Additionally, we repeated GNOM analysis of SAXS profiles measured at indicated concentrations to resolve the discrepancy between zeroextrapolated data and lowest measured concentrations. The new results are incorporated in Figure 1D (main manuscript) and Figure 1—figure supplement 3B.

9. The outcome of the molecular dynamics simulation may be different if the start structure is adjusted, i.e. if the simulation is done with a start model that has the two domains separated. This should be tested to confirm that the SPN-ARR interaction is not affected by the mutations.

For the MD simulations shown in the manuscript, we used the previously published SPN-ARR crystal structure (PDB: 5g4x, Lilja et al., 2017) as a start structure, where the SPNARR interaction was shown experimentally. Separating both domains would be an interesting way to study particular properties of the SPN-ARR interface. However, the biological significance of such a study would be very limited since the SPN- And ARR-domain are connected to each other by a linker region in nature. Nonetheless we were also interested in the outcome of such an experiment and performed the suggested MD simulations.

**Author response image 2. respfig2:** Molecular dynamics simulations of split SPN-ARR domains. (**A**) Cα backbone RMSF calculated for trajectories of the SHANK3(1-346) fragment (PDB: 5G4X), where the SPN- and ARR-domain have been split in silico in the linker region. Both mutants show increased fluctuations within the first 100 residues. (**B**) Cα backbone RMSD determined for the full protein (aa 1-346) with split SPN/ARR domains. Overall, ASD-associated mutants clearly show significantly increased backbone dynamics (**C**) Cα backbone RMSD restricted to the SPN domain of SHANK3 from “split” trajectories. If the two domains are separated, mutations within the SPN domain elicit a strong increase in backbone dynamics compared to the WT, which is absent if the SPN- and ARR-domain are connected by their linker (shown in (**C**)). Additionally, dynamics within the WT SPN domain appear lower if the domains are split. (**D**) Cα backbone RMSD restricted to the ARR domain of SHANK3 from “split” trajectories. Interestingly, under these conditions ASD-associated mutations also induce a distal increase in dynamics of the ARR domain. (**E**) Frame overlays are shown from trajectories where the SPN- and ARR-domain have been split (every fifth frame loaded, overlay (beginning:step:end) = 0:20:20002 resulting in 1000 frames).

We did not observe a significant divergence of both domains throughout the trajectory. Additionally, RMSF analysis of C_α_ atoms again showed increased fluctuations within the first 100 residues for both mutants (Author response image 2). However, we observed an increase of mutant RMSD values compared to the WT (Author response image 2). This effect was more pronounced for the L68P mutant, which showed a steep RMSD increase after ~100ns, potentially reflecting a mutation-induced perturbation of the SPN-ARR domain interface. The R12C mutant also exhibited increased RMSD values but showed a continuous near-linear RMSD increase rather than a spontaneous steep jump. Notably, R12C mutant RMSDs are higher than the WT only after ~500ns and are observed to be even lower than the WT before. Consistent with previous MD simulations, the SPN-domain alone showed identical RMSD patterns as the whole protein with slightly higher absolute values, confirming the SPN-domain as local hotspot of conformational dynamics (Author response image 2). We also found that both mutations were able to increase conformational dynamics also within the decoupled ARRdomain (Author response image 2). Since this increase in ARR-domain dynamics is absent for the WT, it could be that the mutation-carrying SPN-domain influences peptide backbone dynamics of the ARR-domain via non-covalent contacts.

However, we think these results are hard to interpret and should be taken with caution as they are derived from an artificially split system causing potentially non-native behavior and interactions which might not occur in the intact protein. Therefore, we suggest to exclude these results from the main manuscript to prevent confusion of readers. Apart from that, we already demonstrated by CORAL modelling of zero-extrapolated SAXS data that both ASD-associated mutations indeed disrupt the SPN-ARR interface (Figure 1C, main manuscript, see also Essential Revisions #6).

10. The experimental study of disease related missense mutations on protein structure and stability is not new. See for examples doi.org/10.1074/jbc.M210019200 and 10.1371/journal.pone.0186110. Also, in SHANK3 another disease related missense mutation (R536W) has been investigated structurally doi.org/10.1111/jnc.14322. I would like to suggest that lines 299 and 300 are changed or removed to reflect this better. In addition, the "In a proof-of-principle study" in the abstract may be misleading taking this previous work into account.

We are sorry for this misunderstanding. Of course our claim of novelty was not to show that missense mutations affect protein structure and stability. With proof-of-principal study we meant to link structural effects of disease-associated missense mutations in synaptic proteins with protein properties in their native cellular environment. Our study illustrates the value of deep structural analysis of select missense mutations at the level of molecular dynamics. Albeit both mutations investigated here, were located within the same domain, they had distinct effects on SHANK3 folding in vitro and kinetics at the synapse, suggesting that each new mutation needs to be studied individually and data cannot be directly extrapolated or generalized. We replaced the sentence “However, to our knowledge, distinguishable structural perturbations of disease-relevant protein missense variants have not yet been shown experimentally” (lines 299 and 300) with “However, to our knowledge, studies correlating distinguishable structural perturbations of disease-relevant protein missense variants with an altered cellular phenotype are very limited (Post et al., *Nature Comm.,* 2020)”.

11. It would strengthen the discussion if the authors could discuss the cellular readout reported here and by others more in the light of their results on the flexibility and stability of the mutants.

In light of our new FRAP and co-IP results, we have re-written part of the discussion and included our interpretations on possible cellular outcomes of changed SHANK3 structure.